# Equivariant Neural Diffusion for Molecule Generation

**François Cornet**
Technical University of Denmark
`frjc@dtu.dk`

**Grigory Bartosh**
University of Amsterdam
`g.bartosh@uva.nl`

**Mikkel N. Schmidt**
Technical University of Denmark
`mnsc@dtu.dk`

**Christian A. Naesseth**
University of Amsterdam
`c.a.naesseth@uva.nl`

## Abstract

We introduce Equivariant Neural Diffusion (END), a novel diffusion model for molecule generation in 3D that is equivariant to Euclidean transformations. Compared to current state-of-the-art equivariant diffusion models, the key innovation in END lies in its learnable forward process for enhanced generative modelling. Rather than pre-specified, the forward process is parameterized through a time- and data-dependent transformation that is equivariant to rigid transformations. Through a series of experiments on standard molecule generation benchmarks, we demonstrate the competitive performance of END compared to several strong baselines for both unconditional and conditional generation.

## 1   Introduction

The discovery of novel chemical compounds with relevant properties is critical to a number of scientific fields, such as drug discovery and materials design (Merchant et al., 2023). However, due to the large size and complex structure of the chemical space (Ruddigkeit et al., 2012), which combines continuous and discrete features, it is notably difficult to search. Additionally, *ab-initio* Quantum Mechanics (QM) methods for computing target properties are often computationally expensive, preventing brute-force enumeration. While some of these heavy computations can be amortized through learned surrogates, the need for innovative search methods remains, and generative models have recently emerged as a promising avenue (Anstine and Isayev, 2023). Such models can learn complex data distributions, that, in turn, can be sampled from to obtain novel samples similar to the original data. Compared to other data modalities such as images or text, molecules present additional challenges as they have to adhere to strict chemical rules, and obey the symmetries of the 3D space.

Currently, the most promising directions for molecule generation in 3D are either auto-regressive models (Gebauer et al., 2019; 2022; Luo and Ji, 2022; Daigavane et al., 2024) building molecules one atom at a time, or Diffusion Models (DMs) (Hoogeboom et al., 2022; Vignac et al., 2023; Le et al., 2024) that learn to revert a corruption mechanism that transforms the data distribution into noise. As both approaches directly operate in 3D space, they can leverage architectures designed for machine learned force fields (Unke et al., 2021), that were carefully developed to encode the symmetries inherent to the data (Schütt et al., 2017; 2021; Batzner et al., 2022; Batatia et al., 2022).

The success of DMs has not been limited to molecule generation, and promising results have been demonstrated on a variety of other data modalities (Yang et al., 2023). Nevertheless, most existing DMs pre-specify the forward process, forcing the reverse process to comply with it. A recent line of work has sought to overcome that limitation and improve generation by replacing the fixed forward process with a learnable one (Bartosh et al., 2023; Nielsen et al., 2024; Bartosh et al., 2024).

**Contributions** In this paper, we present Equivariant Neural Diffusion (END), a novel diffusion model for molecule generation in 3D that (1) is equivariant to Euclidean transformations, and (2) features a learnable forward process. We demonstrate competitive unconditional molecule generation performance on the QM9 and GEOM-Drugs benchmarks. For conditional generation driven by composition and substructure constraints, our approach exhibits a substantial performance gain compared to existing equivariant diffusion models. Our set of experiments underscores the benefit of a learnable forward process for improved unconditional and conditional molecule generation.

## 2 Background

We begin by establishing the necessary background for generative modeling of geometric graphs. We first introduce the data representation and its inherent symmetries. We then discuss Diffusion Models (DMs), and more specifically the Equivariant Diffusion Model (EDM) (Hoogeboom et al., 2022). Finally, we present the Neural Flow Diffusion Models (NFDM) framework (Bartosh et al., 2024).

### 2.1 Equivariance

**Molecules as geometric graphs in E(3)** We consider geometric graphs embedded in 3-dimensional Euclidean space that represent molecules. Formally, each atomistic system can be described by a tuple $\boldsymbol{x} = (\boldsymbol{r}, \boldsymbol{h})$, where $\boldsymbol{r} = (\boldsymbol{r}_1, ..., \boldsymbol{r}_M) \in \mathbb{R}^{M \times 3}$ form a collection of vectors in 3D representing the coordinates of the atoms, and $\boldsymbol{h} = (\boldsymbol{h}_1, ..., \boldsymbol{h}_M) \in \mathbb{R}^{M \times D}$ are the associated scalar features (e.g. atomic types or charges). When dealing with molecules, we are particularly interested in $E(3)$, the Euclidean group in 3 dimensions, generated by translations, rotations and reflections. Each group element in $E(3)$ can be represented as a combination of a translation vector $\mathbf{t} \in \mathbb{R}^3$ and an orthogonal matrix $\mathbf{R} \in O(3)$ encoding rotation or reflection. While scalar features $\boldsymbol{h}$ remain invariant, coordinates $\boldsymbol{r}$ transform under translation, rotation and reflection as $\mathbf{R}\boldsymbol{r} + \mathbf{t} = (\mathbf{R}\boldsymbol{r}_1 + \mathbf{t}, ..., \mathbf{R}\boldsymbol{r}_M + \mathbf{t})$.

**Equivariant functions** A function $f : \mathcal{X} \to \mathcal{Y}$ is said to be equivariant to the action of a group $G$, or $G$-equivariant, if $g \cdot f(\boldsymbol{x}) = f(g \cdot \boldsymbol{x}), \forall g \in G$. It is said to be $G$-invariant, if $f(\boldsymbol{x}) = f(g \cdot \boldsymbol{x}), \forall g \in G$. In the case of a function $f : (\mathbb{R}^{M \times 3} \times \mathbb{R}^{M \times D}) \to (\mathbb{R}^{M \times 3} \times \mathbb{R}^{M \times D})$ operating on geometric graphs, the function is said to be $E(3)$-equivariant if

$$\mathbf{R}\boldsymbol{y}^{(r)} + \mathbf{t}, \boldsymbol{y}^{(h)} = f\Big(\mathbf{R}\boldsymbol{r} + \mathbf{t}, \boldsymbol{h}\Big), \forall\, \mathbf{R} \in O(3) \text{ and } \mathbf{t} \in \mathbb{R}^3,$$

where $\boldsymbol{y}^{(r)}$ and $\boldsymbol{y}^{(h)}$ denote the output related to $\boldsymbol{r}$ and $\boldsymbol{h}$ respectively. There exists a large variety of graph neural network architectures designed to be equivariant to the Euclidean group (Schütt et al., 2017; 2021; Batzner et al., 2022; Batatia et al., 2022).

**Equivariant distributions** A conditional distribution $p(\boldsymbol{y}|\boldsymbol{x})$ is equivariant to rotations and reflections when $p(\boldsymbol{y}|\boldsymbol{x}) = p(\mathbf{R}\boldsymbol{y}|\mathbf{R}\boldsymbol{x}), \forall\, \mathbf{R} \in O(3)$, while a distribution is said to be invariant when $p(\boldsymbol{x}) = p(\mathbf{R}\boldsymbol{x}), \forall\, \mathbf{R} \in O(3)$. Regarding translation, it is not possible to have a translation-invariant non-zero distribution, as it would require that $p(\boldsymbol{x}) = p(\boldsymbol{x} + \mathbf{t}), \forall \mathbf{t} \in \mathbb{R}^3, \boldsymbol{x} \in \mathbb{R}^{M \times 3}$, which would mean that $p(\boldsymbol{x})$ cannot integrate to 1 (Garcia Satorras et al., 2021). However, a translation-invariant distribution can be constructed in the linear subspace $R$, where the centre of gravity is fixed to $\mathbf{0}$ (i.e. zero CoM subspace): $R = \{\boldsymbol{r} \in \mathbb{R}^{M \times 3} : \frac{1}{M} \sum_{i=1}^{M} \boldsymbol{r}_i = \mathbf{0}\}$ (Xu et al., 2022). As $R$ can be shown to be intrinsically equivalent to $\mathbb{R}^{(M-1) \times 3}$ (Bao et al., 2023), we will consider in what follows that $\boldsymbol{r}$ is defined in $\mathbb{R}^{(M-1) \times 3}$ for ease of notation.

### 2.2 Equivariant Diffusion Models

Diffusion Models (DMs) (Sohl-Dickstein et al., 2015; Ho et al., 2020) are generative models that learn distributions through a hierarchy of latent variables, corresponding to perturbed versions of the data at increasing noise scales. DMs consist of a forward and a reverse (or generative) process. The Equivariant Diffusion Model (EDM) (Hoogeboom et al., 2022) is a particular instance of a DM, where the learned marginal $p_\theta(\boldsymbol{x})$ is made invariant to the action of translations, rotations and reflections by construction. Intuitively, this means that the likelihood of a given molecule under the model does not depend on its orientation.

**Forward process** The forward process perturbs samples from the data distribution, $\boldsymbol{x} \sim q(\boldsymbol{x})$, over time through noise injection, resulting in a trajectory of latent variables $(\boldsymbol{z}_t)_{t \in [0,1]}$, conditional on $\boldsymbol{x}$. The conditional distribution for $(\boldsymbol{z}_t)_{t \in [0,1]}$ given $\boldsymbol{x}$, can be described by an initial distribution $q(\boldsymbol{z}_0|\boldsymbol{x})$ and a Stochastic Differential Equation (SDE),

$$d\{\boldsymbol{z}_t^{(r)}, \boldsymbol{z}_t^{(h)}\} = f(t)\big[\boldsymbol{z}_t^{(r)}, \boldsymbol{z}_t^{(h)}\big]\,dt + g(t)\,d\{\boldsymbol{w}^{(r)}, \boldsymbol{w}^{(h)}\},$$

where the drift $f(t)$ and volatility $g(t)$ are scalar functions of time, and $\boldsymbol{w}^{(r)}$ and $\boldsymbol{w}^{(h)}$ are two independent standard Wiener processes defined in $\mathbb{R}^{(M-1)\times 3}$ and $\mathbb{R}^{M \times D}$ respectively. Specifically, EDM implements the Variance-Preserving SDE (VP-SDE) scheme (Song et al., 2020), with $f(t) = -\frac{1}{2}\beta(t)$ and $g(t) = \sqrt{\beta(t)}$ for a fixed schedule $\beta(t)$. Due to the linearity of the drift term, the conditional marginal distribution is known in closed-form (Särkkä and Solin, 2019), and can be reconstructed as

$$q\big([\boldsymbol{z}_t^{(r)}, \boldsymbol{z}_t^{(h)}] \big| [\boldsymbol{r}, \boldsymbol{h}]\big) = q(\boldsymbol{z}_t^{(r)}|\boldsymbol{r})q(\boldsymbol{z}_t^{(h)}|\boldsymbol{h}) = \mathcal{N}(\boldsymbol{z}_t^{(r)}, |\alpha_t \boldsymbol{r}, \sigma_t^2 \mathbb{I}) \cdot \mathcal{N}(\boldsymbol{z}_t^{(h)}, |\alpha_t \boldsymbol{h}, \sigma_t^2 \mathbb{I}),$$

where $\alpha_t = \exp\left(-\frac{1}{2}\int_0^t \beta(s)\,ds\right)$ and $\sigma_t = 1 - \exp\left(-\frac{1}{2}\int_0^t \beta(s)\,ds\right)$. It evolves from a low-variance Gaussian centered around the data $q(\boldsymbol{z}_0|\boldsymbol{x}) \approx \mathcal{N}(\boldsymbol{z}_0|\boldsymbol{x}, \delta^2\mathbb{I})$ to an uninformative prior distribution (that contains no information about the data distribution), i.e. a unit Gaussian $q(\boldsymbol{z}_1|\boldsymbol{x}) \approx \mathcal{N}(\boldsymbol{z}_1|\boldsymbol{0}, \mathbb{I})$.

**Reverse (generative) process** Starting from the prior $[\boldsymbol{z}_1^{(r)}, \boldsymbol{z}_1^{(h)}] \sim \mathcal{N}(\boldsymbol{z}_t^{(r)}|\boldsymbol{0}, \mathbb{I}) \cdot \mathcal{N}(\boldsymbol{z}_t^{(h)}|\boldsymbol{0}, \mathbb{I})$, samples from $q(\boldsymbol{x})$ can be generated by reversing the forward process. This can be done by following the reverse-time SDE (Anderson, 1982),

$$d\boldsymbol{z}_t = \Big(f(t)\big[\boldsymbol{z}_t^{(r)}, \boldsymbol{z}_t^{(h)}\big] - g^2(t)\big[\nabla_{\boldsymbol{z}_t^{(r)}}\log q(\boldsymbol{z}_t), \nabla_{\boldsymbol{z}_t^{(h)}}\log q(\boldsymbol{z}_t)\big]\Big)dt + g(t)\,d\{\bar{\boldsymbol{w}}^{(r)}, \bar{\boldsymbol{w}}^{(h)}\},$$

where $\bar{\boldsymbol{w}}^{(r)}$ and $\bar{\boldsymbol{w}}^{(h)}$ are independent standard Wiener processes defined in $\mathbb{R}^{(M-1)\times 3}$ and $\mathbb{R}^{M \times D}$, respectively, with time flowing backwards. DMs approximate the reverse process by learning an approximation of the score function $\nabla_{\boldsymbol{z}_t}\log q(\boldsymbol{z}_t)$ parameterized by a neural network $s_\theta(\boldsymbol{z}_t, t)$. With the learned score function $s_\theta(\boldsymbol{z}_t, t)$, a sample $\boldsymbol{z}_0 \sim p_\theta(\boldsymbol{z}_0) \approx q(\boldsymbol{z}_0) \approx q(\boldsymbol{x})$ can be obtained by first sampling from the prior $[\boldsymbol{z}_1^{(r)}, \boldsymbol{z}_1^{(h)}] \sim \mathcal{N}(\boldsymbol{z}_t^{(r)}|\boldsymbol{0}, \mathbb{I}) \cdot \mathcal{N}(\boldsymbol{z}_t^{(h)}|\boldsymbol{0}, \mathbb{I})$, and then simulating the reverse SDE,

$$d\boldsymbol{z}_t = \Big(f(t)\big[\boldsymbol{z}_t^{(r)}, \boldsymbol{z}_t^{(h)}\big] - g^2(t)\big[s_\theta^{(r)}(\boldsymbol{z}_t, t), s_\theta^{(h)}(\boldsymbol{z}_t, t)\big]\Big)dt + g(t)\,d\{\bar{\boldsymbol{w}}^{(r)}, \bar{\boldsymbol{w}}^{(h)}\},$$

where the true score function has been replaced by its approximation $s_\theta(\boldsymbol{z}_t, t)$. In EDM, the approximate score is parameterized through an equivariant function: $s_\theta(\boldsymbol{z}_t, t) = \big[s_\theta^{(r)}(\boldsymbol{z}_t, t), s_\theta^{(h)}(\boldsymbol{z}_t, t)\big]$ such that $s_\theta([\mathbf{R}\boldsymbol{z}_t^{(r)}, \boldsymbol{z}_t^{(h)}], t) = \big[\mathbf{R}s_\theta^{(r)}(\boldsymbol{z}_t, t), s_\theta^{(h)}(\boldsymbol{z}_t, t)\big], \forall\,\mathbf{R} \in O(3)$. Practically, this is realized through the specific parameterization,

$$s_\theta(\boldsymbol{z}_t, t) = \frac{\alpha_t \hat{\boldsymbol{x}}_\theta(\boldsymbol{z}_t, t) - \boldsymbol{z}_t}{\sigma_t^2},$$

where the data point predictor $\hat{\boldsymbol{x}}_\theta$ is implemented by an equivariant neural network.

**Optimization** The data point predictor $\hat{\boldsymbol{x}}_\theta$, or $s_\theta$, is trained by optimizing the denoising score matching loss (Vincent, 2011),

$$\mathcal{L}_{\text{DSM}} = \mathbb{E}_{u(t), q(\boldsymbol{x}, \boldsymbol{z}_t)}\Big[\lambda(t)\big|\big|s_\theta(\boldsymbol{z}_t, t) - \nabla_{\boldsymbol{z}_t}\log q(\boldsymbol{z}_t|\boldsymbol{x})\big|\big|_2^2\Big],$$

where $\lambda(t)$ is a positive weighting function, and $u(t)$ is a uniform distribution over the interval $[0, 1]$.

### 2.3 Neural Flow Diffusion Models

Neural Flow Diffusion Models (NFDM) (Bartosh et al., 2024) are based on the observation that latent variables in DMs, i.e. $\boldsymbol{z}_t$, are conventionally inferred through a pre-specified transformation – as implied by the chosen type of SDE and the noise schedule. This potentially limits the flexibility of the latent space, and makes the learning of the reverse (generative) process more challenging.

**Forward process** In contrast to conventional DMs, NFDM define the forward process implicitly through a learnable transformation $F_\varphi(\boldsymbol{\varepsilon}, t, \boldsymbol{x})$ of injected noise $\boldsymbol{\varepsilon}$, time $t$, and data point $\boldsymbol{x}$. The latent

variables $z_t$ are obtained by transforming noise samples $\varepsilon$, conditional on data point $x$ and time step $t$: $z_t = F_\varphi(\varepsilon, t, x)$. If $F_\varphi$ is differentiable with respect to $\varepsilon$ and $t$, and invertible with respect to $\varepsilon$, then, for fixed $x$ and $\varepsilon$, samples from $q_\varphi(z_t|x)$ can be obtained by solving the following conditional Ordinary Differential Equation (ODE) until time $t$,

$$dz_t = f_\varphi(z_t, t, x)\, dt \ \text{ with } f_\varphi(z_t, t, x) = \left.\frac{\partial F_\varphi(\varepsilon, t, x)}{\partial t}\right|_{\varepsilon = F_\varphi^{-1}(z_t, t, x)}, \quad (1)$$

with $z_0 \sim q(z_0|x)$. While $F_\varphi$ and $q(\varepsilon)$ define the conditional marginal distribution $q_\varphi(z_t|x)$, we need a distribution over the trajectories $(z_t)_{t\in[0,1]}$. NFDM obtain this through the introduction of a conditional SDE starting from $z_0$ and running forward in time. Given access to the ODE in Eq. (1) and the score function $\nabla_{z_t} \log q_\varphi(z_t|x)$, the conditional SDE sharing the same conditional marginal distribution $q_\varphi(z_t|x)$ is given by

$$dz_t = f_\varphi^F(z_t, t, x)\, dt + g_\varphi(t)\, dw \ \text{ with } f_\varphi^F(z_t, t, x) = f_\varphi(z_t, t, x) + \frac{g_\varphi^2(t)}{2}\nabla_{z_t} \log q_\varphi(z_t|x), \quad (2)$$

where the score function of $q_\varphi(z_t|x)$ is $\nabla_{z_t} \log q_\varphi(z_t|x) = \nabla_{z_t}\big[\log q(\varepsilon) + \log|J_F^{-1}|\big]$, with $\varepsilon = F_\varphi^{-1}(z_t, t, x)$, and $J_F^{-1} = \frac{\partial F_\varphi^{-1}(z_t, t, x)}{\partial z_t}$.

**Reverse (generative) process**  A conditional reverse SDE that starts from $z_1 \sim q(z_1)$, runs backward in time, and reverses the conditional forward SDE from Eq. (2) can be defined as

$$dz_t = f_\varphi^B(z_t, t, x) + g_\varphi(t)\, d\bar{w} \ \text{ with } f_\varphi^B(z_t, t, x) = f_\varphi(z_t, t, x) - \frac{g_\varphi^2(t)}{2}\nabla_{z_t} \log q_\varphi(z_t|x). \quad (3)$$

As $x$ is unknown when generating samples, we can rewrite Eq. (3) with the prediction of $x$ instead,

$$dz_t = \hat{f}_{\theta,\varphi}(z_t, t)\, dt + g_\varphi(t)\, d\bar{w}, \ \text{ where } \hat{f}_{\theta,\varphi}(z_t, t) = f_\varphi^B\big(z_t, t, \hat{x}_\theta(z_t, t)\big), \quad (4)$$

where $\hat{x}_\theta$ is a function that predicts the data point $x$. Provided that the reconstruction distribution $q(z_0|x)$ and prior distribution $q(z_1)$ are defined, this fully specifies the reverse (generative) process.

**Optimization**  The forward and reverse processes can be optimized jointly by matching the drift terms of the true and approximate conditional reverse SDEs through the following objective,

$$\mathcal{L}_{\text{NFDM}} = \mathbb{E}_{u(t), q_\varphi(x, z_t)}\Big[\frac{1}{2g_\varphi^2(t)}\big|\big|f_\varphi^B(z_t, t, x) - \hat{f}_{\theta,\varphi}(z_t, t)\big|\big|_2^2\Big], \quad (5)$$

which can be shown to be equivalent to minimizing the Kullback-Leibler divergence between the posterior distributions resulting from the discretization of Eqs. (3) and (4) (Bartosh et al., 2024).

## 3 Equivariant Neural Diffusion

We now introduce Equivariant Neural Diffusion (END), that generalizes the Equivariant Diffusion Model (EDM) (Hoogeboom et al., 2022), by defining the forward process through a learnable transformation. Our approach is a synthesis of NFDM introduced in Section 2.3, and leverages ideas of EDM outlined in Section 2.2 to maintain the desired invariance of the learned marginal distribution $p_{\theta,\varphi}(z_0)$. By providing an equivariant learnable transformation $F_\varphi$ and an equivariant data point predictor $\hat{x}_\theta$, we show that it is possible to obtain a generative model with the desired properties. Finally, we propose a simple yet flexible parameterization meeting these requirements.

### 3.1 Formulation

The key innovation in END lies in its forward process, which is also leveraged in the reverse (generative) process. The forward process is defined through a learnable time- and data-dependent transformation $F_\varphi(\varepsilon, t, x)$, such that the latent $z_t$ transforms covariantly with the injected noise $\varepsilon$ (i.e. a collection of random vectors) and the data point $x$,

$$F_\varphi(\mathbf{R}\varepsilon, t, \mathbf{R}x) = \mathbf{R}F_\varphi(\varepsilon, t, x) = \mathbf{R}z_t, \quad \forall\, \mathbf{R} \in O(3).$$

We then define $\hat{x}_\theta$ as another learnable equivariant function, such that the predicted data point transforms covariantly with the latent variable $z_t$, i.e. $\hat{x}_\theta(\mathbf{R}z_t, t) = \mathbf{R}\hat{x}_\theta(z_t, t)$. Finally, we choose the noise and prior distribution, i.e. $p(\varepsilon)$ and $p(z_1)$, to be invariant to the considered symmetry group.

**Invariance of the learned distribution** With the following choices: (1) $p(z_1)$ an invariant distribution, (2) $F_\varphi$ an equivariant function that satisfies $F_\varphi(\mathbf{R}\varepsilon, t, \mathbf{R}x) = \mathbf{R}F_\varphi(\varepsilon, t, x)$, and (3) $\hat{x}_\theta$ an equivariant function, we have that the learned marginal $p_{\theta,\varphi}(z_0)$ is invariant as desired. This can be shown by demonstrating that the reverse SDE is equivariant. We start by noting that the reverse SDE in END is given by

$$d z_t = \hat{f}_{\theta,\varphi}(z_t, t)\, dt + g_\varphi(t)\, d\bar{w}\,.$$

As the Wiener process is isotropic, this boils down to verifying that the drift term, $\hat{f}_{\theta,\varphi}(z_t, t)$ is equivariant, i.e. $\hat{f}_{\theta,\varphi}(\mathbf{R}z_t, t) = \mathbf{R}\hat{f}_{\theta,\varphi}(z_t, t)$. As the drift is expressed as a sum of two terms, we inspect each of them separately. The first term is

$$f_\varphi\big(z_t, t, \hat{x}_\theta(z_t, t)\big) = \frac{\partial F_\varphi\big(\varepsilon, t, \hat{x}_\theta(z_t, t)\big)}{\partial t}\bigg|_{\varepsilon = F_\varphi^{-1}(z_t, t, \hat{x}_\theta(z_t, t))}.$$

If $F_\varphi$ is equivariant, then so is its time-derivative (see Appendix A.3.1). The same holds for its inverse with respect to $\varepsilon$ (see Appendix A.3.2), such that we have $F_\varphi^{-1}(\mathbf{R}z_t, t, \mathbf{R}x) = \mathbf{R}F_\varphi^{-1}(z_t, t, x) = \mathbf{R}\varepsilon$. We additionally have that $\hat{x}_\theta$ is equivariant, by definition. As the equivariance of $F_\varphi$ implies the equivariance of $q_\varphi$, from the second term of the drift, we observe that, for $y_t = \mathbf{R}z_t$, we have

$$\nabla_{y_t} \log q_\varphi\big(y_t | \hat{x}_\theta(y_t, t)\big) = \mathbf{R}\nabla_{z_t} \log q_\varphi\big(z_t | \hat{x}_\theta(z_t, t)\big), \quad \forall\, \mathbf{R} \in O(3).$$

In summary, in addition to an invariant prior, an equivariant $F_\varphi$ and an equivariant $\hat{x}_\theta$ ensure the equivariance of the reverse process, and hence the invariance of the learned distribution. In Appendix A.3.3, we additionally show that the objective function is invariant, i.e. $\mathcal{L}_{\mathrm{END}}(\mathbf{R}x) = \mathcal{L}_{\mathrm{END}}(x), \forall \mathbf{R} \in O(3)$.

We note that alternative formulations for the drift of the reverse process, $\hat{f}_{\theta,\varphi}$, exist. Most notably, it can be learned directly through an equivariant function, without explicit dependence on $F_\varphi$, while maintaining the desired invariance.

## 3.2 Parameterization

We now introduce a simple parameterization of $F_\varphi$ that meets the requirements outlined above,

$$F_\varphi(\varepsilon, t, x) = \mu_\varphi(x, t) + U_\varphi(x, t)\varepsilon, \tag{6}$$

where, due to the geometric nature of $x$, $U_\varphi(x, t) \in \mathbb{R}^{(M-1) \times 3 \times 3}$ is structured as a block-diagonal matrix where each block is a $3 \times 3$ matrix, ensuring a cheap calculation of the inverse transformation and its Jacobian. This is equivalent to a diagonal parameterization in the case of scalar features.

Similarly to EDM, our parametrization of $F_\varphi$ leads to a conditional marginal $q_\varphi(z_t | x)$ that is a conditional Gaussian with (block-) diagonal covariance, with the notable difference that the mean and covariance are now data- and time-dependent, and learnable through $F_\varphi$,

$$q_\varphi(z_t | x) = \mathcal{N}\big(z_t | \mu_\varphi(x, t), \Sigma_\varphi(x, t)\big), \tag{7}$$

where $\Sigma_\varphi(x, t) = U_\varphi(x, t)U_\varphi^\top(x, t)$, such that $\Sigma_\varphi(x, t)$ is also block-diagonal. As $F_\varphi$ is linear in $\varepsilon$, both $\mu_\varphi$ and $U_\varphi$ must be equivariant functions whose outputs transform covariantly with $x$, in order to ensure the desired equivariance of $F_\varphi$,

$$F_\varphi(\mathbf{R}\varepsilon, t, \mathbf{R}x) = \mu_\varphi(\mathbf{R}x, t) + U_\varphi(\mathbf{R}x, t)\mathbf{R}\varepsilon = \mathbf{R}\mu_\varphi(x, t) + \mathbf{R}\big[U_\varphi(x, t)\varepsilon\big] = \mathbf{R}F_\varphi(\varepsilon, t, x).$$

We can then readily check that the resulting $q_\varphi$ is equivariant, as $\forall\, \mathbf{R} \in O(3)$, we have that

$$q_\varphi(z_t | x) = \mathcal{N}\big(z_t | \mu_\varphi(x, t), \Sigma_\varphi(x, t)\big) = \mathcal{N}\big(\mathbf{R}z_t | \mathbf{R}\mu_\varphi(x, t), \mathbf{R}\Sigma_\varphi(x, t)\mathbf{R}^\top\big) = q_\varphi(\mathbf{R}z_t | \mathbf{R}x).$$

We note that other, and more advanced, parametrizations are possible, e.g. based on normalizing flows with a flow architecture similar to that of Klein et al. (2024).

**Prior and Reconstruction** While not strictly required, it can be advantageous to parameterize the transformation $F_\varphi$ such that the prior and reconstruction losses need not be computed. To do so, we design $F_\varphi(\varepsilon, x, t)$ such that the conditional distribution evolves from a low-variance Gaussian centered around the data, i.e. $q(z_0 | x) \approx \mathcal{N}(z_0 | x, \delta^2 \mathbb{I})$ to an uninformative prior distribution (that

contains no information about the data distribution), i.e. a unit Gaussian $q(\boldsymbol{z}_1|\boldsymbol{x}) \approx \mathcal{N}(\boldsymbol{z}_1|\boldsymbol{0}, \mathbb{I})$. Specifically, we parameterize $F_\varphi$ through the following functions,

$$\mu_\varphi(\boldsymbol{x}, t) = (1-t)\boldsymbol{x} + t(1-t)\bar{\mu}_\varphi(\boldsymbol{x}, t), \tag{8}$$

$$U_\varphi(\boldsymbol{x}, t) = \left(\delta^{1-t}\bar{\sigma}_\varphi(\boldsymbol{x}, t)^{t(1-t)}\right)\mathbb{I} + t(1-t)\bar{U}_\varphi(\boldsymbol{x}, t), \tag{9}$$

which ensure that (i) in $t = 0$, $\mu_\varphi(\boldsymbol{x}, 0) = \boldsymbol{x}$ and $\Sigma_\varphi(\boldsymbol{x}, 0) = \delta^2\mathbb{I}$; (ii) in $t = 1$, $\mu_\varphi(\boldsymbol{x}, 1) = \boldsymbol{0}$ and $\Sigma_\varphi(\boldsymbol{x}, 1) = \mathbb{I}$; while being unconstrained for $t \in ]0, 1[$. We note that this is only one possible parametrization for $F_\varphi$, and that, by adapting $\mu_\varphi$ and $U_\varphi$, richer priors can easily be leveraged – e.g. harmonic prior given a molecular graph or scale-dependent prior (Jing et al., 2023; Irwin et al., 2024).

**Implementation** In practice, $F_\varphi$ is implemented as a neural network with an architecture similar to that of the data point predictor $\hat{\boldsymbol{x}}_\theta(\boldsymbol{z}_t, t)$, but with a specific readout layer that produces $[\bar{\mu}_\varphi(\boldsymbol{x}, t), \bar{\sigma}_\varphi(\boldsymbol{x}, t), \bar{U}_\varphi(\boldsymbol{x}, t)]$. The mean output $\bar{\mu}_\varphi(\boldsymbol{x}, t)$ is similar to that of $\hat{\boldsymbol{x}}_\theta(\boldsymbol{z}_t, t)$. For $U_\varphi(\boldsymbol{x}, t)$, as $\Sigma_\varphi(\boldsymbol{x}, t) = U_\varphi(\boldsymbol{x}, t)U_\varphi^\top(\boldsymbol{x}, t)$ should rotate properly, $\bar{\sigma}_\varphi(\boldsymbol{x}, t)$ is a positive invariant scalar, while $\bar{U}_\varphi(\boldsymbol{x}, t)$ is constructed as a matrix whose columns are vectors that transform covariantly with $\boldsymbol{x}$.

To ease notation, we introduced all notations in the linear subspace $R$, however in practice we work in the ambient space, i.e. $\boldsymbol{r} \in \mathbb{R}^{M \times 3}$. We detail in Appendix A.5.1, how working in ambient space is possible. The training and sampling procedures are detailed in Algorithms 1 and 2 in the appendix.

### 3.3 Conditional Model

While unconditional generation is a required stepping stone, many practical applications require some form of controllability. As other generative models, DMs can model conditional distributions $p(\boldsymbol{x}|\boldsymbol{c})$, where $\boldsymbol{c}$ is a given condition. Different methods exist for sampling from the conditional distribution, e.g. via guidance (Bao et al., 2023; Jung et al., 2024) or twisting (Wu et al., 2024), but the simplest approach consists in training a conditional model on pairs $(\boldsymbol{x}, \boldsymbol{c})$. In such setting, $F_\varphi$ and $\hat{x}_\theta$ simply receive an extra input $\boldsymbol{c}$ representing the conditional information, such that they respectively become $F_\varphi(\boldsymbol{\varepsilon}, t, \boldsymbol{x}, \boldsymbol{c})$, and $\hat{x}_\theta(\boldsymbol{z}_t, t, \boldsymbol{c})$. It is important to note that, compared to conventional DMs, the forward process of END is now also condition-dependent.

## 4 Experiments

In this section, we demonstrate the benefits of END with a comprehensive set of experiments. In Section 4.1, we first display the advantages of END for unconditional generation on 2 standard benchmarks, namely QM9 (Ramakrishnan et al., 2014) and GEOM-DRUGS (Axelrod and Gomez-Bombarelli, 2022). Then, in Section 4.2, we perform conditional generation in 2 distinct settings on QM9. Additional experimental details are provided in Appendix A.6.

**Datasets** The QM9 dataset (Ramakrishnan et al., 2014) contains 134 thousand small- and medium-sized organic molecules with up to 9 heavy atoms, and up to 29 when counting hydrogen atoms. GEOM-DRUGS (Axelrod and Gomez-Bombarelli, 2022) contains 430 thousand medium- and large-sized drug-like molecules with 44 atoms on average, and up to maximum 181 atoms. We use the same data setup as in previous work (Hoogeboom et al., 2022; Xu et al., 2022).

### 4.1 Unconditional Generation

**Task** We sample 10 000 molecules using the stochastic sampling procedure detailed in Algorithm 2. As END is trained in continuous-time, we vary the number of integration steps from 50 to 1000. We repeat each sampling for 3 seeds, and report averages along with standard deviations for each metric.

**Evaluation metrics** We follow previous work (Hoogeboom et al., 2022; Xu et al., 2023), and first evaluate the chemical quality of the generated samples in terms of stability, validity, and uniqueness (in Tables 1, 6 and 9). On QM9, we additionally evaluate how well the model learns the atom and bond types distributions by measuring the total variation between the dataset's and generated distributions, as well as the overall quality of the generated structures via their strain energy, expressed as the energy difference between the generated structure and a relaxed version thereof (in Tables 2 and 6). On GEOM-DRUGS, we additionally compute connectivity, total variation for atom types, and strain energy (in Tables 2 and 9). Connectivity accounts for the fact that validity can easily

Table 1: Stability and validity results on QM9 and GEOM-DRUGS obtained over 10000 samples, with mean/standard deviation across 3 sampling runs. END compares favorably to the baseline across all metrics on both datasets, while offering competitive performance for reduced number of sampling steps. It reaches a performance level similar to that of current SOTA methods. ‡ denotes results obtained by our own experiments.

| | | QM9 | | | | GEOM-DRUGS | | |
| | | Stability (↑) | | Val. / Uniq. (↑) | | Stability (↑) | Val. / Conn. (↑) | |
| Model | Steps | A [%] | M [%] | V [%] | V×U [%] | A [%] | V [%] | V×C [%] |
|---|---|---|---|---|---|---|---|---|
| Data | | 99.0 | 95.2 | 97.7 | 97.7 | 86.5 | 99.0 | 99.0 |
| EDM (Hoogeboom et al., 2022) | 1000 | 98.7 | 82.0 | 91.9 | 90.7 | 81.3 | 92.6 | — |
| EDM-BRIDGE (Wu et al., 2022) | 1000 | 98.8 | 84.6 | 92.0 | 90.7 | 82.4 | 92.8 | — |
| GEOLDM (Xu et al., 2023) | 1000 | $98.9_{\pm.1}$ | $89.4_{\pm.5}$ | $93.8_{\pm.4}$ | $92.7_{\pm.5}$ | 84.4 | 99.3 | $45.8^{\ddagger}$ |
| GEOBFN (Song et al., 2024) | 50 | $98.3_{\pm.1}$ | $85.1_{\pm.5}$ | $92.3_{\pm.4}$ | $90.7_{\pm.3}$ | 75.1 | 91.7 | |
| | 100 | $98.6_{\pm.1}$ | $87.2_{\pm.3}$ | $93.0_{\pm.3}$ | $91.5_{\pm.3}$ | 78.9 | 93.1 | — |
| | 500 | $98.8_{\pm.8}$ | $88.4_{\pm.2}$ | $93.4_{\pm.2}$ | $91.8_{\pm.2}$ | 81.4 | 93.5 | |
| | 1000 | $99.1_{\pm.1}$ | $90.9_{\pm.2}$ | $95.3_{\pm.1}$ | $93.0_{\pm.1}$ | 85.6 | 92.1 | |
| EDM* | 50 | $97.6_{\pm.0}$ | $77.6_{\pm.5}$ | $90.2_{\pm.2}$ | $89.2_{\pm.2}$ | $84.7_{\pm.0}$ | $93.6_{\pm.2}$ | $46.6_{\pm.3}$ |
| | 100 | $98.1_{\pm.0}$ | $81.9_{\pm.4}$ | $92.1_{\pm.2}$ | $90.9_{\pm.2}$ | $85.2_{\pm.1}$ | $93.8_{\pm.3}$ | $56.2_{\pm.4}$ |
| | 250 | $98.3_{\pm.0}$ | $84.3_{\pm.1}$ | $93.2_{\pm.4}$ | $91.7_{\pm.3}$ | $85.4_{\pm.0}$ | $94.2_{\pm.1}$ | $61.4_{\pm.6}$ |
| | 500 | $98.4_{\pm.0}$ | $85.2_{\pm.5}$ | $93.5_{\pm.2}$ | $92.2_{\pm.3}$ | $85.4_{\pm.0}$ | $94.3_{\pm.2}$ | $63.4_{\pm.1}$ |
| | 1000 | $98.4_{\pm.0}$ | $85.3_{\pm.3}$ | $93.5_{\pm.1}$ | $91.9_{\pm.1}$ | $85.3_{\pm.1}$ | $94.4_{\pm.1}$ | $64.2_{\pm.6}$ |
| **END** | 50 | $98.6_{\pm.0}$ | $84.6_{\pm.1}$ | $92.7_{\pm.1}$ | $91.4_{\pm.1}$ | $87.1_{\pm.1}$ | $84.6_{\pm.5}$ | $66.0_{\pm.4}$ |
| | 100 | $98.8_{\pm.0}$ | $87.4_{\pm.2}$ | $94.1_{\pm.0}$ | $92.3_{\pm.2}$ | $87.2_{\pm.1}$ | $87.0_{\pm.2}$ | $73.7_{\pm.4}$ |
| | 250 | $98.9_{\pm.1}$ | $88.8_{\pm.5}$ | $94.7_{\pm.2}$ | $92.6_{\pm.1}$ | $87.1_{\pm.1}$ | $88.5_{\pm.2}$ | $77.4_{\pm.4}$ |
| | 500 | $98.9_{\pm.0}$ | $88.8_{\pm.4}$ | $94.8_{\pm.2}$ | $92.8_{\pm.2}$ | $87.0_{\pm.0}$ | $88.8_{\pm.3}$ | $78.6_{\pm.3}$ |
| | 1000 | $98.9_{\pm.0}$ | $89.1_{\pm.1}$ | $94.8_{\pm.1}$ | $92.6_{\pm.2}$ | $87.0_{\pm.0}$ | $89.2_{\pm.3}$ | $79.4_{\pm.4}$ |

be increased by generating several disconnected fragments (where only the largest counts towards validity), the total variation ensures that the model properly samples all atom types, whereas the strain energy evaluates the generated geometries. On both datasets, we also report additional drug-related metrics as per Qiang et al. (2023) (in Tables 7 and 10). More details about evaluation metrics are provided in Appendix A.6.1.

**Baselines** We compare END to several relevant baselines from the literature: the original EDM (Hoogeboom et al., 2022); EDM-BRIDGE (Wu et al., 2022), an improved version of EDM that adds a physics-inspired force guidance in the reverse process; GEOLDM (Xu et al., 2023), an equivariant latent DM; and GEOBFN (Song et al., 2024), a geometric version of Bayesian Flow Networks (Graves et al., 2024). A detailed discussion about related work can be found in Section 5.

**Ablations of END** In addition to baselines from the literature, we compare different ablated versions of END. As the key component to our method is the learnable forward process, the logical ablation is whether to include a learnable forward (=END), or not (=EDM). To ensure a fair comparison and that any difference in performance does not stem from an increase in learnable parameters, an architectural change or the training paradigm, we implement our own EDM (Hoogeboom et al., 2022), denoted EDM*. It features the exact same architecture as END, the same amount of learnable parameters (i.e. through a deeper $\hat{x}_\theta$), and is similarly trained in continuous-time. Additionally, we provide EDM* + $\gamma$, similar to EDM* but with a learned SNR (Kingma et al., 2021) for each data modality (i.e. atomic types and coordinates), and END ($\mu_\varphi$ only), an ablated version of END where only the mean is learned whereas the standard deviation of the conditional marginal is pre-specified and derived from the noise schedule of EDM. Table 5 provides an overview of the compared models.

**Results on QM9** Our main results on the QM9 dataset are summarized in the left part of Tables 1 and 2, where END is shown to significantly outperform the baseline and reach a level of performance similar to current SOTA methods, GEOLDM (Xu et al., 2023) and GEOBFN (Song et al., 2024), across stability and validity metrics. The ablation study in Appendix A.2.1 clearly reveals the benefits of a learnable forward. On the one hand, the two variants of END are shown to outperform (or be on par with) all baselines across all metrics in Table 6 (in particular, in terms of the more challenging

Table 2: Additional results on QM9 and GEOM-DRUGS obtained over 10000 samples, with mean/standard deviation across 3 sampling runs. END matches better the training distributions, and generates less strained structures than baselines. ‡ denotes results obtained by our own experiments.

| | | QM9 | | | GEOM-DRUGS | |
| | | TV ($\downarrow$) | | Strain En. ($\downarrow$) | TV ($\downarrow$) | Strain En. ($\downarrow$) |
| Model | Steps | A [$10^{-2}$] | B [$10^{-3}$] | $\Delta$E [kcal/mol] | A [$10^{-2}$] | $\Delta$E [kcal/mol] |
|---|---|---|---|---|---|---|
| Data | | | | 7.7 | | 15.8 |
| GEOLDM (Xu et al., 2023) | 1000 | $1.6^{\ddagger}$ | $1.3^{\ddagger}$ | $10.4^{\ddagger}$ | $10.6^{\ddagger}$ | $133.5^{\ddagger}$ |
| EDM* | 50 | $4.6_{\pm.1}$ | $1.7_{\pm.5}$ | $16.4_{\pm.2}$ | $10.5_{\pm.1}$ | $134.2_{\pm1.5}$ |
| | 100 | $3.5_{\pm.1}$ | $1.4_{\pm.3}$ | $13.5_{\pm.1}$ | $8.0_{\pm.1}$ | $110.9_{\pm0.3}$ |
| | 250 | $2.8_{\pm.2}$ | $1.3_{\pm.4}$ | $12.3_{\pm.4}$ | $6.7_{\pm.1}$ | $98.9_{\pm0.3}$ |
| | 500 | $2.6_{\pm.2}$ | $1.3_{\pm.4}$ | $11.7_{\pm.1}$ | $6.4_{\pm.1}$ | $95.3_{\pm0.1}$ |
| | 1000 | $2.5_{\pm.1}$ | $1.4_{\pm.4}$ | $11.3_{\pm.1}$ | $6.2_{\pm.0}$ | $92.9_{\pm1.1}$ |
| **END** | 50 | $1.4_{\pm.1}$ | $1.9_{\pm.4}$ | $13.1_{\pm.1}$ | $5.9_{\pm.1}$ | $86.3_{\pm.6}$ |
| | 100 | $1.1_{\pm.0}$ | $1.5_{\pm.2}$ | $11.1_{\pm.1}$ | $4.5_{\pm.1}$ | $67.9_{\pm.9}$ |
| | 250 | $1.0_{\pm.0}$ | $0.5_{\pm.1}$ | $10.3_{\pm.1}$ | $3.5_{\pm.0}$ | $58.9_{\pm.1}$ |
| | 500 | $0.9_{\pm.0}$ | $0.7_{\pm.1}$ | $10.0_{\pm.1}$ | $3.3_{\pm.0}$ | $56.4_{\pm.8}$ |
| | 1000 | $0.9_{\pm.2}$ | $1.0_{\pm.1}$ | $9.7_{\pm.2}$ | $3.0_{\pm.0}$ | $55.0_{\pm.5}$ |

molecule stability), and be in better agreement with the data distribution in Table 7 (except for QED which is captured perfectly by all methods). On the other hand, we observe that, with as few as 100 integration steps, END is capable of generating samples that are qualitatively better than those generated by the simpler baselines in 1000 steps. A few illustrative QM9-like molecules generated by END are displayed in Fig. 1.

**Results on GEOM-DRUGS** Our main results on the more realistic GEOM-DRUGS dataset are presented in the right part of Tables 1 and 2, where we observe that END is competitive against the compared baselines in terms of atom stability, while being slightly subpar in terms of validity. However, when accounting for connectivity (via the V×C metric), we observe that END does outperform the baseline with an increased sucess rate of around 20% on average, as well as better than the SOTA method GEOLDM (Xu et al., 2023) by a large margin. The generated molecules also better follow the atom types distribution of the dataset – as per the lower total variation, and feature better geometries than those generated by competitors – as implied by the lower strain energy. As for QM9, the ablation study provided in Appendix A.2.3 illustrates the clear improvement provided by a learnable forward process compared to a fixed one. In this more challenging setting, only learning the mean function yields a slight decrease in performance across the metrics reported in Table 9, and a similar agreement with the dataset in terms of the drug-related metrics in Table 10, compared to the full model. Furthermore, while each sampling step is currently slower than EDM* (Table 11), the improved sampling efficiency afforded by END (in terms of integration steps) allows practical time gains on this more complex dataset. As a concrete example, samples obtained with END with only 100 steps are qualitatively better those generated by EDM* in 1000 integration steps, amounting to a 3x time cut. Examples of GEOM-DRUGS-like molecules generated by END are provided in Fig. 1.

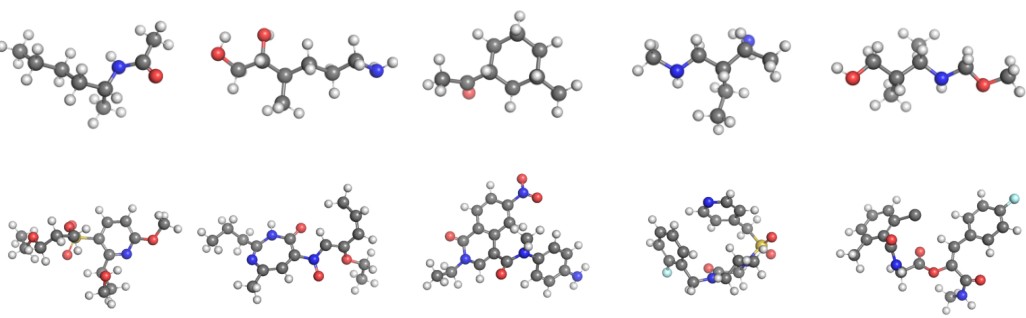

Figure 1: Representative samples generated by END on QM9 (top), and GEOM-DRUGS (bottom).

## 4.2 Conditional Generation

**Dataset and Setup** We perform our experiments on the QM9 dataset, on 2 different tasks: composition-conditioned and substructure-conditioned generation. Both tasks allow for direct validation with ground-truth properties without requiring expensive QM calculations, or approximations with surrogate models. In each case, we train a conditional diffusion model as described in Section 3.3, i.e. where $F_\varphi$ and $\hat{x}_\theta$ are provided with an extra input corresponding to the condition. Additional details are provided in Appendix A.6.4.

**Task 1: composition-conditioned generation** The model is tasked to generate a compound with a predefined composition, i.e. structural isomers of a given formula. The condition is specified as a vector $\boldsymbol{c} = (c_1, ..., c_D) \in \mathbb{Z}^D$, where $c_d$ denotes the number of atoms of type $d$ that the sample should contain. To evaluate the model, we generate 10 samples per target formula, and compute the proportion of samples that match the provided composition. Our results are provided in Table 3, where we observe that CEND significantly outperforms the baseline, and offers nearly fully controllable composition generation. Additionally, reducing the number of sampling steps has a very limited impact on the controllability. Finally, we perform an ablation whose results are presented in Table 8, where fixing the standard deviation is shown to lead to a small decrease in performance, with respect to the full model while remaining significantly better than the baseline with fixed forward.

**Task 2: substructure-conditioned generation** We adopt a setup similar to that of Bao et al. (2023), and train a conditional END, where the condition is a molecular fingerprint encoding structural information about the molecule. A fingerprint is a binary vector $\boldsymbol{c} = (c_1, ..., c_F) \in \{0, 1\}^F$, where $c_f$ is set to 1 if substructure $f$ is present in the molecule, or to 0 if not. Fingerprints are obtained using OPENBABEL (O'Boyle et al., 2011). We evaluate the ability of the compared models to leverage the provided structural information, by sampling conditionally on fingerprints obtained from the test set. We then compute the Tanimoto similarity between the fingerprints yielded by generated molecules and the fingerprints provided as conditional inputs. We compare CEND to EEGSDE (Bao et al., 2023), an improved version of EDM (Hoogeboom et al., 2022), that performs conditional generation by combining a conditional diffusion model and regressor guidance. Our results are presented in Table 4, along with a handful of samples in Fig. 2, where CEND is shown to offer better controllability than the compared baselines, as highlighted by the higher similarity.

## 5 Related Work

The main approaches to molecule generation in 3D are autoregressive models (Gebauer et al., 2019; Simm et al., 2020; Gebauer et al., 2022; Luo and Ji, 2022; Daigavane et al., 2024), flow-based models (Garcia Satorras et al., 2021), and diffusion models (Hoogeboom et al., 2022; Igashov et al., 2024). A notable exception to the geometric graph representation of 3D molecules are voxels (Skalic et al., 2019; Ragoza et al., 2022; O Pinheiro

Table 3: On composition-conditioned generation, CEND offers nearly perfect composition controllability. Matching refers to the % of samples featuring the prompted composition.

| Model | Steps | Matching [%] (↑) |
|---|---|---|
| | 50 | $69.6_{\pm 0.6}$ |
| | 100 | $73.0_{\pm 0.6}$ |
| CEDM* | 250 | $74.1_{\pm 1.4}$ |
| | 500 | $76.2_{\pm 0.6}$ |
| | 1000 | $75.5_{\pm 0.5}$ |
| | 50 | $89.2_{\pm 0.8}$ |
| | 100 | $90.1_{\pm 1.0}$ |
| **CEND** | 250 | $91.2_{\pm 0.8}$ |
| | 500 | $91.5_{\pm 0.8}$ |
| | 1000 | $91.0_{\pm 0.9}$ |

Table 4: On substructure-conditioned generation, CEND shows competitive performance, surpassing EEGSDE that uses an additional property predictor. [†]Results are borrowed from Bao et al. (2023).

| Model | Steps | Tanimoto Sim. (↑) |
|---|---|---|
| CEDM[†] | 1000 | $.671_{\pm .004}$ |
| EEGSDE[†] | 1000 | $.750_{\pm .003}$ |
| | 50 | $.601_{\pm .000}$ |
| | 100 | $.640_{\pm .002}$ |
| CEDM* | 250 | $.663_{\pm .002}$ |
| | 500 | $.669_{\pm .001}$ |
| | 1000 | $.673_{\pm .002}$ |
| | 50 | $.783_{\pm .001}$ |
| | 100 | $.807_{\pm .001}$ |
| **CEND** | 250 | $.819_{\pm .001}$ |
| | 500 | $.825_{\pm .001}$ |
| | 1000 | $.828_{\pm .001}$ |

Figure 2: Excerpt of substructure-conditioned samples, where CEND matches the provided substructure better (in terms of compositions and local patterns).



et al., 2024), from which the 3D graph is extracted via some post-processing procedure. Recently, several works have shown that leveraging 2D connectivity information can lead to improved results (Peng et al., 2023; Vignac et al., 2023; Le et al., 2024). While not incompatible with END, we perform our experiments without modeling that auxiliary information, and therefore do not compare to these approaches directly. Other frameworks have also been tailored to molecule generation, such as Flow Matching (Lipman et al., 2022; Song et al., 2023; Irwin et al., 2024) or Bayesian Flow Networks (Graves et al., 2024; Song et al., 2024), also showing promises for accelerated sampling.

In the realm of diffusion models for molecules, EDM-BRIDGE (Wu et al., 2022) and EEGSDE (Bao et al., 2023) extend upon a continuous-time formulation of EDM (Hoogeboom et al., 2022), as END also does. Based on the observation that there exists an infinity of processes mapping from prior to target distributions, EDM-BRIDGE constructs one such process that incorporates some prior knowledge, i.e. part of the drift term is a physically-inspired force term. END can be seen as a generalization of EDM-BRIDGE, where the forward drift term is now learned instead of pre-specified. Through experiments, we show that a learnable forward performs better than a fixed one, even when the latter is physics-inspired. EEGSDE specifically targets conditional generation by combining (1) a conditional model score model, (2) a method similar to classifier-guidance (requiring the training of an auxiliary model). In CEND, we instead only learn a conditional model. Finally, GEOLDM (Xu et al., 2023) is a latent diffusion model that performs diffusion in the latent space of an equivariant Variational Auto-Encoder (VAE), and it can be seen as a particular case of END, where $F_\varphi(\varepsilon, t, x) = \alpha_t E_\varphi(x) + \epsilon \sigma_t$, with $E_\varphi(x)$ denoting the (time-independent) encoder of the VAE.

## 6 Conclusion

In this work, we have presented Equivariant Neural Diffusion (END), a novel diffusion model that is equivariant to Euclidean transformations. The key innovation in END lies in the forward process that is specified by a learnable data- and time-dependent transformation. Experimental results demonstrate the benefits of our method. In the unconditional setting, we show that END yields competitive generative performance across two different benchmarks. In the conditional setting, END offers improved controllability, when conditioning on composition and substructure. Finally, as a by-product of the introduced learnable forward, we also find the sampling efficiency (in terms of integration steps) to be improved, while that property is not actively optimized for in the design of the parameterization nor in the training procedure.

Avenues for future work are numerous. In particular, further leveraging the flexible framework of NFDM (Bartosh et al., 2024) to constrain the generative trajectories, e.g. to be straight and enable even faster sampling, modelling bond information, or extending the conditional setting to other types of conditioning information, e.g. other point cloud or target property, are all promising directions.

**Limitations**   From a computational perspective, END is currently slower to train and sample from compared to a baseline with fixed forward with identical number of learnable parameters. Even if convergence is similarly fast, each training (resp. sampling) step incurs a relative $\approx 2.5$x (resp. $\approx 3$x). However, END usually requires much fewer number of function evaluations to achieve comparable (or better) accuracy, and alternative (and more efficient) parameterizations of the reverse process exist. In particular, the drift $\hat{f}_{\theta,\varphi}$ could be learned without direct dependence on $f_\varphi$, thereby leading to an improved training time and, more importantly, a very limited overhead with respect to vanilla DMs for sampling. In terms of scalability, END suffers from limitations similar to concurrent approaches. It operates on fully-connected graphs – preventing its scaling to very large graphs, and models categorical features through an arbitrary continuous relaxation – potentially suboptimal and scaling linearly with the number of chemical elements in the modeled dataset. Encodings that scale more gracefully, such as that of Analog Bits (Chen et al., 2023) (logarithmic) or GeoLDM (Xu et al., 2023) (learned low-dimensional), are good candidates to better deal with discrete features within END. In terms of data, the presented findings are limited to organic molecules and the metrics, while widely used in the community, also have some evident limitations. To fully assess the practical interest of the generated molecules, thorough validation with QM simulations would be required.

**Broader Impact**   Generative models for molecules have the potential to accelerate *in-silico* discovery and design of drugs or materials. This work proposes an instance of such model. As any generative model, it also comes with potential dangers, as this could be misused for designing e.g. chemicals with adversarial properties.

## Acknowledgments and Disclosure of Funding

FC acknowledges financial support from the Independent Research Fund Denmark with project DELIGHT (Grant No. 0217-00326B).

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

# A Appendices

## A.1 Compared models

In Table 5, we detail all the compared diffusion models in terms of their transformation $F_\varphi$.

Table 5: Compared models.

| | $F_\varphi(\varepsilon, t, \boldsymbol{x})$ | Comment |
|---|---|---|
| EDM (Hoogeboom et al., 2022) / EDM* | $\alpha_t \boldsymbol{x} + \sigma_t \varepsilon$ | $\alpha_t = \exp\left(-\frac{1}{2}\int_0^t \beta(s)\,ds\right)$ $\sigma_t = 1 - \exp\left(-\frac{1}{2}\int_0^t \beta(s)\,ds\right)$ |
| GEOLDM (Xu et al., 2023) | $\alpha_t E_\varphi(\boldsymbol{x}) + \sigma_t \varepsilon$ | $\alpha_t$ and $\sigma_t$ similar to EDM $p(\boldsymbol{x}|\boldsymbol{z}_0) = \mathcal{N}\left(\boldsymbol{x}|D_\varphi(\boldsymbol{z}_0), \delta^2 \mathbb{I}\right)$ |
| EDM* + $\gamma_\varphi$ | $\alpha_\varphi(t)\boldsymbol{x} + \sigma_\varphi(t)\varepsilon$ | learned $\gamma_\varphi$ with 2 outputs ($\boldsymbol{r}$ and $\boldsymbol{h}$) |
| END ($\mu_\varphi$ only) | $\mu_\varphi(\boldsymbol{x}, t) + \sigma_t \varepsilon$ | $\sigma_t$ similar to EDM |
| END | $\mu_\varphi(\boldsymbol{x}, t) + U_\varphi(\boldsymbol{x}, t)\varepsilon$ | as introduced in Eq. (6) |

## A.2 Additional results

### A.2.1 Ablation on unconditional QM9

Table 6: Main results of the ablation study on QM9. Metrics are obtained over 10000 samples, with mean/standard deviation across 3 sampling runs. The two variants of END compare favorably to baselines across metrics, while offering competitive performance for reduced number of sampling steps.

| | | Stability (↑) | | Validity / Uniqueness (↑) | | TV (↓) | | Strain En. (↓) |
|---|---|---|---|---|---|---|---|---|
| Model | Steps | A [%] | M [%] | V [%] | V×U [%] | A $[10^{-2}]$ | B $[10^{-3}]$ | $\Delta$E [kcal/mol] |
| Data | | 99.0 | 95.2 | 97.7 | 97.7 | | | 7.7 |
| EDM* | 50 | $97.6_{\pm.0}$ | $77.6_{\pm.5}$ | $90.2_{\pm.2}$ | $89.2_{\pm.2}$ | $4.6_{\pm.1}$ | $1.7_{\pm.5}$ | $16.4_{\pm.2}$ |
| | 100 | $98.1_{\pm.0}$ | $81.9_{\pm.4}$ | $92.1_{\pm.2}$ | $90.9_{\pm.2}$ | $3.5_{\pm.1}$ | $1.4_{\pm.3}$ | $13.5_{\pm.1}$ |
| | 250 | $98.3_{\pm.0}$ | $84.3_{\pm.1}$ | $93.2_{\pm.4}$ | $91.7_{\pm.3}$ | $2.8_{\pm.2}$ | $1.3_{\pm.4}$ | $12.3_{\pm.4}$ |
| | 500 | $98.4_{\pm.0}$ | $85.2_{\pm.5}$ | $93.5_{\pm.2}$ | $92.2_{\pm.3}$ | $2.6_{\pm.2}$ | $1.3_{\pm.4}$ | $11.7_{\pm.1}$ |
| | 1000 | $98.4_{\pm.0}$ | $85.3_{\pm.3}$ | $93.5_{\pm.1}$ | $91.9_{\pm.1}$ | $2.5_{\pm.1}$ | $1.4_{\pm.4}$ | $11.3_{\pm.1}$ |
| EDM* + $\gamma_\varphi$ | 50 | $97.7_{\pm.0}$ | $77.4_{\pm.3}$ | $91.1_{\pm.4}$ | $90.2_{\pm.4}$ | $4.3_{\pm.1}$ | $1.5_{\pm.2}$ | $15.5_{\pm.2}$ |
| | 100 | $98.2_{\pm.0}$ | $82.6_{\pm.2}$ | $92.9_{\pm.2}$ | $91.6_{\pm.2}$ | $3.2_{\pm.1}$ | $1.2_{\pm.2}$ | $12.8_{\pm.1}$ |
| | 250 | $98.5_{\pm.0}$ | $85.3_{\pm.3}$ | $93.9_{\pm.1}$ | $92.4_{\pm.1}$ | $2.5_{\pm.1}$ | $1.0_{\pm.1}$ | $11.3_{\pm.0}$ |
| | 500 | $98.5_{\pm.1}$ | $86.1_{\pm.4}$ | $94.1_{\pm.2}$ | $92.5_{\pm.2}$ | $2.2_{\pm.1}$ | $1.0_{\pm.3}$ | $11.1_{\pm.1}$ |
| | 1000 | $98.5_{\pm.0}$ | $86.1_{\pm.3}$ | $94.1_{\pm.2}$ | $92.4_{\pm.2}$ | $2.1_{\pm.1}$ | $1.1_{\pm.1}$ | $10.8_{\pm.1}$ |
| **END ($\mu_\varphi$ only)** | 50 | $98.5_{\pm.0}$ | $83.9_{\pm.2}$ | $95.2_{\pm.2}$ | $93.8_{\pm.3}$ | $1.4_{\pm.1}$ | $1.9_{\pm.4}$ | $13.1_{\pm.1}$ |
| | 100 | $98.7_{\pm.0}$ | $87.0_{\pm.3}$ | $95.5_{\pm.2}$ | $93.6_{\pm.2}$ | $1.1_{\pm.0}$ | $1.5_{\pm.2}$ | $11.1_{\pm.1}$ |
| | 250 | $98.9_{\pm.0}$ | $89.0_{\pm.2}$ | $95.8_{\pm.2}$ | $93.8_{\pm.2}$ | $1.0_{\pm.0}$ | $0.5_{\pm.1}$ | $10.3_{\pm.1}$ |
| | 500 | $98.9_{\pm.0}$ | $88.6_{\pm.2}$ | $95.6_{\pm.1}$ | $93.5_{\pm.1}$ | $0.9_{\pm.0}$ | $0.7_{\pm.1}$ | $10.0_{\pm.1}$ |
| | 1000 | $98.9_{\pm.0}$ | $89.2_{\pm.3}$ | $95.6_{\pm.1}$ | $93.5_{\pm.1}$ | $0.9_{\pm.2}$ | $1.0_{\pm.1}$ | $9.7_{\pm.2}$ |
| **END** | 50 | $98.6_{\pm.0}$ | $84.6_{\pm.1}$ | $92.7_{\pm.1}$ | $91.4_{\pm.1}$ | $1.5_{\pm.1}$ | $1.9_{\pm.4}$ | $12.1_{\pm.3}$ |
| | 100 | $98.8_{\pm.0}$ | $87.4_{\pm.2}$ | $94.1_{\pm.0}$ | $92.3_{\pm.2}$ | $1.3_{\pm.0}$ | $1.8_{\pm.3}$ | $10.6_{\pm.2}$ |
| | 250 | $98.9_{\pm.1}$ | $88.8_{\pm.5}$ | $94.7_{\pm.2}$ | $92.6_{\pm.1}$ | $1.2_{\pm.1}$ | $0.8_{\pm.2}$ | $9.6_{\pm.2}$ |
| | 500 | $98.9_{\pm.0}$ | $88.8_{\pm.4}$ | $94.8_{\pm.2}$ | $92.8_{\pm.2}$ | $1.2_{\pm.1}$ | $0.8_{\pm.5}$ | $9.5_{\pm.1}$ |
| | 1000 | $98.9_{\pm.0}$ | $89.1_{\pm.1}$ | $94.8_{\pm.1}$ | $92.6_{\pm.2}$ | $1.2_{\pm.1}$ | $0.8_{\pm.5}$ | $9.3_{\pm.1}$ |

Table 7: Additional results of the ablation study with metrics from HIERDIFF (Qiang et al., 2023) and MMD (Gretton et al., 2012; Daigavane et al., 2024) on QM9. The two variants of END shows better, or on par, agreement with the true data distribution compared to the baselines.

| Model | Steps | SA ($\uparrow$) | QED ($\uparrow$) | logP ($\uparrow$) | MW | MMD ($\downarrow$) [$10^{-1}$] |
|---|---|---|---|---|---|---|
| Data | | 0.626 | 0.462 | 0.339 | 122.7 | 7.7 |
| EDM* | 50 | $0.609_{\pm.001}$ | $0.456_{\pm.000}$ | $-0.049_{\pm.008}$ | $125.7_{\pm.0}$ | $1.91_{\pm.03}$ |
| | 100 | $0.613_{\pm.001}$ | $0.458_{\pm.000}$ | $0.049_{\pm.003}$ | $124.9_{\pm.1}$ | $1.67_{\pm.02}$ |
| | 250 | $0.617_{\pm.001}$ | $0.461_{\pm.001}$ | $0.107_{\pm.013}$ | $124.4_{\pm.0}$ | $1.52_{\pm.02}$ |
| | 500 | $0.618_{\pm.001}$ | $0.462_{\pm.000}$ | $0.124_{\pm.006}$ | $124.2_{\pm.1}$ | $1.50_{\pm.04}$ |
| | 1000 | $0.619_{\pm.000}$ | $0.462_{\pm.001}$ | $0.135_{\pm.007}$ | $124.2_{\pm.0}$ | $1.51_{\pm.02}$ |
| EDM* + $\gamma_\varphi$ | 50 | $0.612_{\pm.000}$ | $0.454_{\pm.001}$ | $-0.053_{\pm.003}$ | $125.3_{\pm.0}$ | $2.04_{\pm.02}$ |
| | 100 | $0.616_{\pm.001}$ | $0.459_{\pm.000}$ | $0.049_{\pm.010}$ | $124.6_{\pm.0}$ | $1.66_{\pm.01}$ |
| | 250 | $0.620_{\pm.001}$ | $0.461_{\pm.000}$ | $0.124_{\pm.011}$ | $124.1_{\pm.0}$ | $1.54_{\pm.02}$ |
| | 500 | $0.620_{\pm.001}$ | $0.461_{\pm.000}$ | $0.144_{\pm.005}$ | $124.0_{\pm.1}$ | $1.50_{\pm.02}$ |
| | 1000 | $0.622_{\pm.000}$ | $0.462_{\pm.001}$ | $0.162_{\pm.008}$ | $123.9_{\pm.1}$ | $1.45_{\pm.03}$ |
| END ($\mu_\varphi$ only) | 50 | $0.606_{\pm.001}$ | $0.456_{\pm.001}$ | $0.096_{\pm.003}$ | $123.8_{\pm.1}$ | $2.80_{\pm.06}$ |
| | 100 | $0.615_{\pm.001}$ | $0.458_{\pm.000}$ | $0.157_{\pm.002}$ | $123.6_{\pm.1}$ | $1.97_{\pm.05}$ |
| | 250 | $0.622_{\pm.002}$ | $0.460_{\pm.000}$ | $0.198_{\pm.005}$ | $123.3_{\pm.1}$ | $1.48_{\pm.03}$ |
| | 500 | $0.626_{\pm.001}$ | $0.462_{\pm.000}$ | $0.219_{\pm.006}$ | $123.1_{\pm.1}$ | $1.36_{\pm.03}$ |
| | 1000 | $0.627_{\pm.001}$ | $0.462_{\pm.001}$ | $0.225_{\pm.011}$ | $123.0_{\pm.0}$ | $1.36_{\pm.02}$ |
| END | 50 | $0.602_{\pm.001}$ | $0.456_{\pm.001}$ | $0.074_{\pm.012}$ | $123.7_{\pm.1}$ | $1.91_{\pm.00}$ |
| | 100 | $0.613_{\pm.001}$ | $0.459_{\pm.000}$ | $0.125_{\pm.010}$ | $123.3_{\pm.1}$ | $1.63_{\pm.02}$ |
| | 250 | $0.620_{\pm.001}$ | $0.461_{\pm.000}$ | $0.164_{\pm.002}$ | $123.1_{\pm.1}$ | $1.44_{\pm.04}$ |
| | 500 | $0.622_{\pm.001}$ | $0.462_{\pm.000}$ | $0.193_{\pm.008}$ | $123.2_{\pm.0}$ | $1.41_{\pm.01}$ |
| | 1000 | $0.623_{\pm.001}$ | $0.462_{\pm.001}$ | $0.198_{\pm.013}$ | $123.0_{\pm.0}$ | $1.37_{\pm.04}$ |

#### A.2.2 Ablation on composition-conditioned QM9

Table 8: Ablation on composition-conditioned generation. The two versions CEND display better controllability. Fixing the standard deviation leads to a small decrease in performance, with respect to the full model while remaining significantly better than the baseline with fixed forward.

| Model | Steps | Matching [%] ($\uparrow$) |
|---|---|---|
| CEDM* | 50 | $69.6_{\pm 0.6}$ |
| | 100 | $73.0_{\pm 0.6}$ |
| | 250 | $74.1_{\pm 1.4}$ |
| | 500 | $76.2_{\pm 0.6}$ |
| | 1000 | $75.5_{\pm 0.5}$ |
| **CEND** ($\mu_\varphi$ **only**) | 50 | $75.7_{\pm 0.4}$ |
| | 100 | $79.9_{\pm 0.4}$ |
| | 250 | $82.7_{\pm 0.5}$ |
| | 500 | $83.0_{\pm 0.8}$ |
| | 1000 | $83.5_{\pm 0.6}$ |
| **CEND** | 50 | $89.2_{\pm 0.8}$ |
| | 100 | $90.1_{\pm 1.0}$ |
| | 250 | $91.2_{\pm 0.8}$ |
| | 500 | $91.5_{\pm 0.8}$ |
| | 1000 | $91.0_{\pm 0.9}$ |

#### A.2.3 Ablations on GEOM-DRUGS

Table 9: Main results of the ablation study on GEOM-DRUGS. Metrics are obtained over 10000 samples, with mean/standard deviation across 3 sampling runs. Most notably, END generates more connected samples, and less strained structures. Fixing the standard deviation leads to a slight decrease in performance.

| Model | Steps | Stability ($\uparrow$) A [%] | Val. / Conn. ($\uparrow$) V [%] | V$\times$C [%] | TV ($\downarrow$) A $[10^{-2}]$ | Strain En. ($\downarrow$) $\Delta$E [kcal/mol] |
|---|---|---|---|---|---|---|
| Data | | 86.5 | 99.0 | 99.0 | | 15.8 |
| EDM* | 50 | $84.7_{\pm .0}$ | $93.6_{\pm .2}$ | $46.6_{\pm .3}$ | $10.5_{\pm .1}$ | $134.2_{\pm 1.5}$ |
| | 100 | $85.2_{\pm .1}$ | $93.8_{\pm .3}$ | $56.2_{\pm .4}$ | $8.0_{\pm .1}$ | $110.9_{\pm .3}$ |
| | 250 | $85.4_{\pm .0}$ | $94.2_{\pm .1}$ | $61.4_{\pm .6}$ | $6.7_{\pm .1}$ | $98.9_{\pm .3}$ |
| | 500 | $85.4_{\pm .0}$ | $94.3_{\pm .2}$ | $63.4_{\pm .1}$ | $6.4_{\pm .1}$ | $95.3_{\pm .1}$ |
| | 1000 | $85.3_{\pm .1}$ | $94.4_{\pm .1}$ | $64.2_{\pm .6}$ | $6.2_{\pm .0}$ | $92.9_{\pm 1.1}$ |
| **END** ($\mu_\varphi$ only) | 50 | $85.6_{\pm .1}$ | $87.8_{\pm .2}$ | $66.0_{\pm .4}$ | $7.9_{\pm .0}$ | $105.6_{\pm 1.0}$ |
| | 100 | $85.8_{\pm .1}$ | $89.9_{\pm .1}$ | $73.7_{\pm .4}$ | $6.1_{\pm .1}$ | $85.5_{\pm 0.5}$ |
| | 250 | $85.7_{\pm .1}$ | $91.2_{\pm .2}$ | $77.4_{\pm .4}$ | $5.0_{\pm .1}$ | $74.5_{\pm 1.5}$ |
| | 500 | $85.8_{\pm .1}$ | $91.6_{\pm .1}$ | $78.6_{\pm .3}$ | $4.8_{\pm .1}$ | $72.3_{\pm 1.1}$ |
| | 1000 | $85.8_{\pm .1}$ | $91.8_{\pm .1}$ | $79.4_{\pm .4}$ | $4.6_{\pm .0}$ | $71.0_{\pm 0.6}$ |
| **END** | 50 | $87.1_{\pm .1}$ | $84.6_{\pm .5}$ | $68.6_{\pm .4}$ | $5.9_{\pm .1}$ | $86.3_{\pm .6}$ |
| | 100 | $87.2_{\pm .1}$ | $87.0_{\pm .2}$ | $76.7_{\pm .5}$ | $4.5_{\pm .1}$ | $67.9_{\pm .9}$ |
| | 250 | $87.1_{\pm .1}$ | $88.5_{\pm .2}$ | $80.7_{\pm .6}$ | $3.5_{\pm .0}$ | $58.9_{\pm .1}$ |
| | 500 | $87.0_{\pm .0}$ | $88.8_{\pm .3}$ | $81.7_{\pm .4}$ | $3.3_{\pm .0}$ | $56.4_{\pm .8}$ |
| | 1000 | $87.0_{\pm .0}$ | $89.2_{\pm .3}$ | $82.5_{\pm .3}$ | $3.0_{\pm .0}$ | $55.0_{\pm .5}$ |

Table 10: Additional results of the ablation study with metrics from HIERDIFF (Qiang et al., 2023) on GEOM-DRUGS. The two variants of END shows better agreement with the true data distribution, compared to the baseline with fixed forward.

| Model | Steps | SA (↑) | QED (↑) | logP (↑) | MW |
|---|---|---|---|---|---|
| Data | | 0.832 | 0.672 | 2.985 | 360.0 |
| EDM* | 50 | $0.590_{\pm.001}$ | $0.480_{\pm.001}$ | $1.105_{\pm.012}$ | $354.2_{\pm0.9}$ |
| | 100 | $0.614_{\pm.002}$ | $0.534_{\pm.001}$ | $1.482_{\pm.022}$ | $352.3_{\pm0.4}$ |
| | 250 | $0.630_{\pm.001}$ | $0.559_{\pm.003}$ | $1.716_{\pm.009}$ | $351.4_{\pm0.4}$ |
| | 500 | $0.637_{\pm.001}$ | $0.570_{\pm.003}$ | $1.794_{\pm.023}$ | $350.7_{\pm0.3}$ |
| | 1000 | $0.641_{\pm.002}$ | $0.574_{\pm.004}$ | $1.831_{\pm.013}$ | $350.5_{\pm1.4}$ |
| **END** ($\mu_\varphi$ only) | 50 | $0.634_{\pm.001}$ | $0.526_{\pm.001}$ | $1.435_{\pm.007}$ | $352.4_{\pm0.6}$ |
| | 100 | $0.664_{\pm.002}$ | $0.568_{\pm.001}$ | $1.792_{\pm.020}$ | $351.3_{\pm0.5}$ |
| | 250 | $0.681_{\pm.000}$ | $0.591_{\pm.002}$ | $1.996_{\pm.014}$ | $351.1_{\pm0.6}$ |
| | 500 | $0.687_{\pm.000}$ | $0.596_{\pm.002}$ | $2.059_{\pm.015}$ | $350.6_{\pm0.7}$ |
| | 1000 | $0.690_{\pm.001}$ | $0.602_{\pm.001}$ | $2.093_{\pm.019}$ | $351.6_{\pm0.4}$ |
| **END** | 50 | $0.621_{\pm.003}$ | $0.487_{\pm.002}$ | $0.939_{\pm.019}$ | $352.0_{\pm1.4}$ |
| | 100 | $0.660_{\pm.001}$ | $0.550_{\pm.002}$ | $1.530_{\pm.010}$ | $351.6_{\pm1.9}$ |
| | 250 | $0.685_{\pm.001}$ | $0.578_{\pm.002}$ | $1.832_{\pm.009}$ | $351.4_{\pm1.3}$ |
| | 500 | $0.695_{\pm.002}$ | $0.586_{\pm.004}$ | $1.945_{\pm.012}$ | $352.1_{\pm1.6}$ |
| | 1000 | $0.698_{\pm.002}$ | $0.590_{\pm.002}$ | $2.009_{\pm.009}$ | $352.0_{\pm0.6}$ |

### A.3 Equivariance / invariance proofs

#### A.3.1 Time-derivative of an O(3)-equivariant function

Let $f : \mathcal{X} \times [0,1] \to \mathcal{Y}$ be a function that is equivariant to actions of the group $O(3)$, such that $\boldsymbol{R} \cdot f(\boldsymbol{x}, t) = f(\boldsymbol{R} \cdot \boldsymbol{x}, t), \forall \boldsymbol{R} \in O(3)$.

**Proof sketch**   We need to show that $\frac{\partial}{\partial t}\big(f(\boldsymbol{R} \cdot \boldsymbol{x}, t)\big) = \boldsymbol{R} \cdot \frac{\partial}{\partial t}\big(f(\boldsymbol{x}, t)\big), \forall \boldsymbol{R} \in O(3)$ and $\forall \boldsymbol{x} \in \mathcal{X}$.

$$
\begin{aligned}
\frac{\partial}{\partial t}\big(f(\boldsymbol{R} \cdot \boldsymbol{x}, t)\big) &= \frac{\partial}{\partial t}\big(\boldsymbol{R} \cdot f(\boldsymbol{x}, t)\big), \\
&= \boldsymbol{R} \cdot \frac{\partial}{\partial t}\big(f(\boldsymbol{x}, t)\big),
\end{aligned}
$$

where the last equality follows by linearity.

#### A.3.2 Inverse of an O(3)-equivariant function

Let $f : \mathcal{X} \to \mathcal{Y}$ be a function that (1) is equivariant to the action of the group $O(3)$, and (2) admits an inverse $f^{-1} : \mathcal{Y} \to \mathcal{X}$, then $f^{-1}$ is also equivariant to the action of $O(3)$.

**Proof sketch**   We need to show that $f^{-1}\big(\boldsymbol{R} \cdot \boldsymbol{y}\big) = \boldsymbol{R} \cdot f^{-1}(\boldsymbol{y}), \forall \boldsymbol{R} \in O(3)$ and $\forall \boldsymbol{y} \in \mathcal{Y}$.

Since $f$ is invertible, we have that to any $\boldsymbol{y} \in \mathcal{Y}$ corresponds a unique $\boldsymbol{x} \in \mathcal{X}$, such that $\boldsymbol{y} = f(\boldsymbol{x})$, and that $f^{-1}(\boldsymbol{y}) = f^{-1}(f(\boldsymbol{x})) = \boldsymbol{x}$. As $f$ is equivariant to the action of $O(3)$, we have that $\boldsymbol{R} \cdot f(\boldsymbol{x}) = f(\boldsymbol{R} \cdot \boldsymbol{x}), \forall \boldsymbol{R} \in O(3)$:

$$
\begin{aligned}
f^{-1}\big(\boldsymbol{R} \cdot \boldsymbol{y}\big) &= f^{-1}\big(\boldsymbol{R} \cdot f(\boldsymbol{x})\big), \\
&= f^{-1}\big(f(\boldsymbol{R} \cdot \boldsymbol{x})\big), \\
&= \boldsymbol{R} \cdot \boldsymbol{x}, \\
&= \boldsymbol{R} \cdot f^{-1}(\boldsymbol{y}).
\end{aligned}
$$

#### A.3.3 O(3)-invariance of the objective function

In this section, we show that the objective function is invariant under the action of $O(3)$: $\mathcal{L}_{\text{END}}(\mathbf{R}\boldsymbol{x}) = \mathcal{L}_{\text{END}}(\boldsymbol{x}), \forall \boldsymbol{R} \in O(3)$, provided that $F_\varphi$ and $\hat{x}_\theta$ are equivariant.

$$\mathcal{L}_{\text{END}}(\mathbf{R}\boldsymbol{x}) = \mathbb{E}_{u(t),q_\varphi(\boldsymbol{z}_t|\mathbf{R}\boldsymbol{x})q(\mathbf{R}\boldsymbol{x})}\Big[\frac{1}{2g_\varphi^2(t)}\big|\big|f_\varphi(\boldsymbol{z}_t,t,\mathbf{R}\boldsymbol{x}) - \frac{g_\varphi^2(t)}{2}\nabla_{\boldsymbol{z}_t}\log q_\varphi(\boldsymbol{z}_t|\mathbf{R}\boldsymbol{x})$$

$$- f_\varphi(\boldsymbol{z}_t,t,\hat{\boldsymbol{x}}_\theta(\boldsymbol{z}_t,t)) + \frac{g_\varphi^2(t)}{2}\nabla_{\boldsymbol{z}_t}\log q_\varphi(\boldsymbol{z}_t|\hat{\boldsymbol{x}}_\theta(\boldsymbol{z}_t,t))\big|\big|_2^2\Big],$$

$$= \mathbb{E}_{u(t),q_\varphi(\boldsymbol{z}_t|\mathbf{R}\boldsymbol{x})q(\mathbf{R}\boldsymbol{x})}\Big[\frac{1}{2g_\varphi^2(t)}\big|\big|f_\varphi(\mathbf{R}\mathbf{R}^{-1}\boldsymbol{z}_t,t,\mathbf{R}\boldsymbol{x}) - \frac{g_\varphi^2(t)}{2}\nabla_{\boldsymbol{z}_t}\log q_\varphi(\mathbf{R}\mathbf{R}^{-1}\boldsymbol{z}_t|\mathbf{R}\boldsymbol{x})$$

$$- f_\varphi(\mathbf{R}\mathbf{R}^{-1}\boldsymbol{z}_t,t,\hat{\boldsymbol{x}}_\theta(\mathbf{R}\mathbf{R}^{-1}\boldsymbol{z}_t,t)) + \frac{g_\varphi^2(t)}{2}\nabla_{\boldsymbol{z}_t}\log q_\varphi(\mathbf{R}\mathbf{R}^{-1}\boldsymbol{z}_t|\hat{\boldsymbol{x}}_\theta(\mathbf{R}\mathbf{R}^{-1}\boldsymbol{z}_t,t))\big|\big|_2^2\Big],$$

$$= \mathbb{E}_{u(t),q_\varphi(\boldsymbol{z}_t|\mathbf{R}\boldsymbol{x})q(\mathbf{R}\boldsymbol{x})}\Big[\frac{1}{2g_\varphi^2(t)}\big|\big|\mathbf{R}f_\varphi(\mathbf{R}^{-1}\boldsymbol{z}_t,t,\boldsymbol{x}) - \frac{g_\varphi^2(t)}{2}\nabla_{\boldsymbol{z}_t}\log q_\varphi(\mathbf{R}^{-1}\boldsymbol{z}_t|\boldsymbol{x})$$

$$- \mathbf{R}f_\varphi(\mathbf{R}^{-1}\boldsymbol{z}_t,t,\hat{\boldsymbol{x}}_\theta(\mathbf{R}^{-1}\boldsymbol{z}_t,t)) + \frac{g_\varphi^2(t)}{2}\nabla_{\boldsymbol{z}_t}\log q_\varphi(\mathbf{R}^{-1}\boldsymbol{z}_t|\hat{\boldsymbol{x}}_\theta(\mathbf{R}^{-1}\boldsymbol{z}_t,t))\big|\big|_2^2\Big],$$

$$= \mathbb{E}_{u(t),q_\varphi(\mathbf{R}\boldsymbol{y}_t|\mathbf{R}\boldsymbol{x})q(\mathbf{R}\boldsymbol{x})}\Big[\frac{1}{2g_\varphi^2(t)}\big|\big|\mathbf{R}f_\varphi(\boldsymbol{y}_t,t,\boldsymbol{x}) - \frac{g_\varphi^2(t)}{2}\mathbf{R}\nabla_{\boldsymbol{y}_t}\log q_\varphi(\boldsymbol{y}_t|\boldsymbol{x})$$

$$- \mathbf{R}f_\varphi(\boldsymbol{y}_t,t,\hat{\boldsymbol{x}}_\theta(\boldsymbol{y}_t,t)) + \frac{g_\varphi^2(t)}{2}\mathbf{R}\nabla_{\boldsymbol{y}_t}\log q_\varphi(\boldsymbol{y}_t|\hat{\boldsymbol{x}}_\theta(\boldsymbol{y}_t,t))\big|\big|_2^2\Big],$$

$$= \mathbb{E}_{u(t),q_\varphi(\boldsymbol{y}_t|\boldsymbol{x})q(\boldsymbol{x})}\Big[\frac{1}{2g_\varphi^2(t)}\big|\big|f_\varphi(\boldsymbol{y}_t,t,\boldsymbol{x}) - \frac{g_\varphi^2(t)}{2}\nabla_{\boldsymbol{y}_t}\log q_\varphi(\boldsymbol{y}_t|\boldsymbol{x})$$

$$- f_\varphi(\boldsymbol{y}_t,t,\hat{\boldsymbol{x}}_\theta(\boldsymbol{y}_t,t)) + \frac{g_\varphi^2(t)}{2}\nabla_{\boldsymbol{y}_t}\log q_\varphi(\boldsymbol{y}_t|\hat{\boldsymbol{x}}_\theta(\boldsymbol{y}_t,t))\big|\big|_2^2\Big],$$

$$= \mathcal{L}_{\text{END}}(\boldsymbol{x}).$$

The first equality is obtained by replacing $\boldsymbol{x}$ by $\mathbf{R}\boldsymbol{x}$ in the definition of the objective function Eq. (5). The second is obtained by multiplying by $\mathbf{R}\mathbf{R}^{-1} = \mathbb{I}$. The third equality by leveraging that $f_\varphi$, $q_\varphi$ and $\hat{x}_\theta$ are equivariant. We then perform a change of variable $\boldsymbol{y}_t = \mathbf{R}^{-1}\boldsymbol{z}_t$. As rotation does preserve distances, we obtain the last equality.

## A.4 Algorithms

---

**Algorithm 1** Training algorithm of END

---

**Require:** $q(\boldsymbol{x})$, $F_\varphi$, $\hat{\boldsymbol{x}}_\theta$
  **for** training iterations **do**
      $\boldsymbol{x} \sim q(\boldsymbol{x})$, $t \sim u(t)$, $\boldsymbol{\varepsilon} \sim p(\boldsymbol{\varepsilon})$
      $\boldsymbol{z}_t \leftarrow \mu_\varphi(\boldsymbol{x}, t) + U_\varphi(\boldsymbol{x}, t)\boldsymbol{\varepsilon}$
      $\mathcal{L} = \frac{1}{2g_\varphi^2(t)} \left|\left| f_\varphi^B(\boldsymbol{z}_t, t, \boldsymbol{x}) - \hat{f}_{\theta,\varphi}(\boldsymbol{z}_t, t) \right|\right|_2^2$
      Gradient step on $\theta$ and $\varphi$
  **end for**

---

---

**Algorithm 2** Stochastic sampling from END

---

**Require:** $F_\varphi$, $\hat{\boldsymbol{x}}_\theta$, integration steps $T$, empirical distribution of number of atoms $p(N)$
  $\Delta t = \frac{1}{T}$
  $N \sim p(N)$
  $\boldsymbol{z}_1 \sim p(\boldsymbol{z}_1)$
  **for** $t = 1, ..., \frac{1}{T}$ **do**
      $\bar{\boldsymbol{w}} \sim \mathcal{N}(\boldsymbol{0}, \mathbb{I})$
      $\boldsymbol{z}_{t-\Delta t} \leftarrow \boldsymbol{z}_t - \hat{f}_{\theta,\varphi}(\boldsymbol{z}_t, t)\Delta t + g_\varphi(t)\bar{\boldsymbol{w}}\sqrt{\Delta t}$
  **end for**
  $\boldsymbol{x} \sim p(\boldsymbol{x}|\boldsymbol{z}_0)$

---

## A.5 Details about $F_\varphi$

### A.5.1 Working in ambient space

In this section, we omit the invariant features, and consider that $\boldsymbol{x}, \boldsymbol{z}_t, \boldsymbol{\varepsilon} \in \mathbb{R}^{M \times d}$ are all collections of vectors.

In opposition to the notations introduced in Section 3, we would like $F_\varphi$ and $\hat{x}_\theta$ (which ultimately are neural networks) to operate in ambient space directly, i.e. on $\boldsymbol{z}_t, \boldsymbol{x}, \boldsymbol{\varepsilon} \in R$ with $R = \{\boldsymbol{v} \in \mathbb{R}^{M \times d} : \frac{1}{M} \sum_{i=1}^{M} \boldsymbol{v}_i = \boldsymbol{0}\}$, instead of $\mathbb{R}^{(M-1) \times d}$ as initially presented. To do so, we use the results from Garcia Satorras et al. (2021) that showed that the Jacobian of the transformation $F_\varphi$ can be computed in ambient space, for $\boldsymbol{z}_t$ and $\boldsymbol{\varepsilon}$ that live in the linear subspace $R$, provided that the transformation $F_\varphi$ (invertible with respect to $\boldsymbol{\varepsilon}$) leaves the center of mass of $\boldsymbol{\varepsilon}$ unchanged.

Considering flat representations of $\boldsymbol{z}_t, \boldsymbol{x}, \boldsymbol{\varepsilon} \in \mathbb{R}^{M \cdot d}$, the parameterization of $F_\varphi$ introduced in Eqs. (6), (8) and (9) can be adapted as follows to achieve such property

$$F_\varphi(\boldsymbol{\varepsilon}, t, \boldsymbol{x}) = \mu_\varphi(\boldsymbol{x}, t) + U_\varphi(\boldsymbol{x}, t)\boldsymbol{\varepsilon}, \tag{10}$$

$$\mu_\varphi(\boldsymbol{x}, t) = (\mathbb{I}_{M \cdot d} - \frac{1}{M} TT^\top)\tilde{\mu}_\varphi(\boldsymbol{x}, t) \tag{11}$$

$$U_\varphi(\boldsymbol{x}, t) = (\mathbb{I}_{M \cdot d} - \frac{1}{M} TT^\top)\tilde{U}_\varphi(\boldsymbol{x}, t) + \frac{1}{M} TT^\top, \tag{12}$$

where $T \in \mathbb{R}^{M \cdot d \times d} = [\mathbb{I}_d, \mathbb{I}_d, ...]^\top$, and $\frac{1}{M} TT^\top$ corresponds to the linear operator computing the center of mass. In Eq. (11), the unconstrained mean output $\tilde{\mu}_\varphi(\boldsymbol{x}, t)$ is simply projected onto the $0$-CoM subspace, thereby inducing no translation. In Eq. (12), the unconstrained output of $\tilde{U}_\varphi(\boldsymbol{x}, t)$ is first projected onto the $0$-CoM subspace, before being translated back to the initial center of mass.

With this adapted formulation, we need to be able to (1) compute the latent variable $\boldsymbol{z}_t$ given $\boldsymbol{\varepsilon}$, (2) evaluate the Jacobian of the transformation $F_\varphi$, and (3) evaluate the inverse transformation $F_\varphi^{-1}$.

**Computing $\boldsymbol{z}_t$ from $\boldsymbol{\varepsilon}$** Given that $\boldsymbol{\varepsilon} \in R$, obtaining $\boldsymbol{z}_t \in R$ from $\boldsymbol{\varepsilon}$ simply amounts to (1) computing $\tilde{\boldsymbol{z}}_t = \tilde{\mu}_\varphi(\boldsymbol{x}, t) + \tilde{U}_\varphi(\boldsymbol{x}, t)\boldsymbol{\varepsilon}$, and then (2) removing the center of mass from $\tilde{\boldsymbol{z}}_t$ – the second term in Eq. (12) induces no translation as $\boldsymbol{\varepsilon} \in R$.

**Computing $|J_{F_\varphi}|$ and $F_\varphi^{-1}$** In what follows, we shorten the notation, and denote $U_\varphi(\boldsymbol{x}, t)$ by $U$ and $\tilde{U}_\varphi(\boldsymbol{x}, t)$ by $\tilde{U}$. To leverage known identities, we start by reorganizing Eq. (12), as

$$U = \tilde{U} + \frac{1}{M} TT^\top(\mathbb{I}_{M \cdot d} - \tilde{U}). \tag{13}$$

The Jacobian of $F_\varphi$ is given by the determinant of $U$, and the latter can be derived by leveraging the Matrix Determinant Lemma,

$$\det U = \det \tilde{U} \cdot \det \left(\mathbb{I}_d + \frac{1}{M} T^\top(\mathbb{I}_{M \cdot d} - \tilde{U})\tilde{U}^{-1}T\right), \tag{14}$$

$$= \det \tilde{U} \cdot \det \left(\mathbb{I}_d + \frac{1}{M} T^\top(\tilde{U}^{-1} - \mathbb{I}_{M \cdot d})T\right), \tag{15}$$

$$= \det \tilde{U} \cdot \det \left(\frac{1}{M} \sum_{m=1}^{M} (\tilde{U}^m)^{-1}\right), \tag{16}$$

$$= \det \tilde{U} \cdot \det V, \tag{17}$$

$$= \prod_{m=1}^{M} \det \tilde{U}^m \cdot \det V, \tag{18}$$

where $\tilde{U}^m$ denotes the $m$-th $d \times d$-block in $\tilde{U}$, and $V = \frac{1}{M} \sum_{m=1}^{M} (\tilde{U}^m)^{-1}$ is a $d \times d$-matrix. Computing the Jacobian therefore amounts to compute the inverse of $M$ $d \times d$-matrices and the determinant of $M + 1$ $d \times d$-matrices. In practice, $d = 3$ such that all computations can be performed in closed-form.

Regarding $F_\varphi^{-1}$, we do not need to evaluate the inverse transformation itself, but instead evaluate $\varepsilon$ given $z_t$,

$$\varepsilon = U^{-1}\big(z_t - \mu(x,t)\big).$$

The inverse $U^{-1}$ can be obtained via the Woodbury matrix identity,

$$
\begin{aligned}
U^{-1} &= \big(\tilde{U} + \tfrac{1}{M}TT^\top(\mathbb{I}_{M\cdot d} - \tilde{U})\big)^{-1}, \\
&= \tilde{U}^{-1} - \tfrac{1}{M}\tilde{U}^{-1}TV^{-1}T^\top(\tilde{U}^{-1} - \mathbb{I}_{M\cdot d}), \\
&= \tilde{U}^{-1}(\mathbb{I}_{M\cdot d} - C),
\end{aligned}
$$

where $V = \tfrac{1}{M}\sum_{m=1}^{M}(\tilde{U}^m)^{-1}$ as previously defined in Eq. (18), and $C = \tfrac{1}{M}TV^{-1}T^\top(\tilde{U}^{-1} - \mathbb{I}_{M\cdot d})$. Given the specific structure of $C$, the computation of $\varepsilon$ can be simplified to

$$
\begin{aligned}
\varepsilon &= U^{-1}\bar{z}_t, \\
&= \tilde{U}^{-1}(\mathbb{I} - C)\bar{z}_t, \\
&= \tilde{U}^{-1}\big[\bar{z}_t - c\big],
\end{aligned}
$$

where $\bar{z} = \big(z_t - \mu(x,t)\big)$ and $c = C\bar{z}_t$ acts as a translation operator. We note that computing the inverse transformation requires to invert $M + 1$ $d \times d$-matrices, but as $d = 3$ in practice, all computations can be performed in closed-form.

### A.5.2 Invariant features

For simplicity, we omitted in Section 3 and Appendix A.5.1 that molecules are described as tuples: $x = (r, h)$, as only $r$ transform under Euclidean transformations. For the invariant features $h$, we use the following parameterization

$$\mu_\varphi^{(h)}(x,t) = (1-t)h + t(1-t)\bar{\mu}_\varphi^{(h)}(x,t), \tag{19}$$

$$\sigma_\varphi^{(h)}(x,t) = \delta^{1-t}\bar{\sigma}_\varphi(x,t)^{t(1-t)}. \tag{20}$$

which ensures that $\mu_\varphi^{(h)}(x,0) = h$ and $\sigma_\varphi^{(h)}(x,0) = \delta$; whereas $\mu_\varphi^{(h)}(x,1) = 0$ and $\sigma_\varphi^{(h)}(x,1) = \mathbb{I}$.

**Implementation**    As described in the main text in Section 3.2, $F_\varphi$ is implemented as a neural network with an architecture similar to that of the data point predictor $\hat{x}_\theta(z_t, t)$, but with a specific readout layer that produces the outputs related to $r$ ($[\bar{\mu}_\varphi(x,t), \bar{\sigma}_\varphi, \bar{U}_\varphi(x,t)]$). Additionally, it produces $\bar{\mu}_\varphi^{(h)}(x,t)$ and $\bar{\sigma}_\varphi^{(h)}(x,t)$ as invariant outputs.

**Inverse transformation**    The logarithm of the determinant of the inverse transformation $\log|J_F^{-1}|$ writes

$$\log|J_F^{-1}| = -\log|J_F| = -\underbrace{\sum_{i=1}^{M\times D}\sigma_\varphi^{(h),i}(x,t)}_{\text{invariant features}} - \underbrace{\sum_{i=m}^{M}\log\big|\det(U_\varphi^m(x,t))\big| - \log\det|V|}_{\text{vectorial features}}, \tag{21}$$

where $V$ is defined as in Eq. (18).

### A.6 Experimental details

In addition to the details provided in this section, we release a public code repository with our implementation of END.

### A.6.1 Evaluation metrics

In this section, we describe the metrics employed to evaluate the different models:

- **Stability:** An atom is deemed stable if it has a charge of 0, whereas a molecule is stable if all its atoms have 0 charge. We reuse the lookup table from Hoogeboom et al. (2022) to infer bond types from pairwise distances.

- **Validity and Connectivity:** Validity corresponds to the percentage of samples that can be parsed and sanitized by `rdkit` (Landrum et al., 2013), after inference of the bonds using the lookup table mechanism (Hoogeboom et al., 2022). It should be noted that the metric does not penalize fragmented samples as long as each individual fragment appears valid. This can be problematic when running evaluation on larger compounds such as those in GEOM-DRUGS, as models tend to generated disconnected structures. To account for that, we also report Connectivity, which simply check that a valid molecule is composed of exactly one fragment.

- **Uniqueness:** Uniqueness is expressed as the proportion of samples that are valid and have a unique SMILES string (Weininger, 1988) among all the generated samples. On GEOM-DRUGS, we do not report Uniqueness as all generated samples appear unique (as per previous work).

- **Total variation:** The total variation is computed as the MAE between the (discrete) marginal obtained on the training data and on the generated samples. For bond types on QM9, we compute the ground truth and generated distributions using the lookup table mechanism (Hoogeboom et al., 2022).

- **Strain Energy:** The strain energy is expressed as the difference in energy between generated structures and a relaxed version thereof obtained as per `rdkit`'s MMFF (Landrum et al., 2013). From the generated samples, we infer `rdkit mol` objects using OPENBABEL (O'Boyle et al., 2011). We only evaluate the strain energy of valid and connected samples.

- **SA, QED, $\log P$ and MW:** SA denotes the "Synthetic Accessibility Score", which is a rule-based scoring function that evaluates the complexity of synthesizing a structure by organic reactions (Ertl and Schuffenhauer, 2009). We normalized its values between 0 and 1, with 0 being "difficult to synthesize" and 1 "easy to synthesize". QED denotes "Quantitative Estimation of Drug-likeness". $\log P$ denotes the octanol-water partition coefficient. MW denotes the molecular weight.

  We employ the `rdkit`'s implementation of all metrics. To do so, we convert the generated samples to `rdkit mol` objects using OPENBABEL (O'Boyle et al., 2011). We then only evaluate the different metrics for valid and connected samples.

- **MMD:** On QM9, we follow the procedure of Daigavane et al. (2024), and compute the MMD Gretton et al. (2012) between true and generated pairwise distances distributions for the 10 most common bonds in the dataset: ["C-H:1.0", "C-C:1.0", "C-O:1.0", "C-N:1.0", "H-N:1.0", "C-O:2.0", "C-N:1.5", "H-O:1.0", "C-C:1.5", "C-N:2.0"].

### A.6.2 Architecture

Our forward transformation $F_\varphi$ and data point predictor $\hat{x}_\theta$ share a common neural network architecture that we detail here. The architecture is similar to that of EQCAT (Le et al., 2022), and updates a collection of invariant and equivariant features for each node in the graph. We choose that architecture because it allows for an easy construction of $\bar{U}_\varphi(\boldsymbol{x}, t)$ by linear projection of the final equivariant layer.

We follow previous work (Hoogeboom et al., 2022) and consider fully-connected graphs. We initially featurize pairwise distances through Gaussian Radial Basis functions, with dataset-specific cutoff taken large enough to ensure full connectivity. In opposition to Hoogeboom et al. (2022), we do not update positions in the message-passing phase, but instead obtain the positions prediction through a

linear projection of the final equivariant hidden states. The predictions for the invariant features are obtained by reading out the final invariant hidden states.

**Optimization**  For all model variants, we employ Adam with a learning rate of $10^{-4}$. We perform gradient clipping (norm) with a value of 10 on QM9, and a value of 1 on GEOM-DRUGS.

### A.6.3   Unconditional generation

We reuse the data setup from previous work (Hoogeboom et al., 2022; Xu et al., 2023).

**QM9**  On QM9, we use 10 layers of message passing for EDM*, while the variants of END feature 5 layers of message-passing in $F_\varphi$ and 5 layers in $\hat{x}_\theta$. For all models, we use 256 invariant and 256 equivariant hidden features, along with an RBF expansion of dimension 64 with a cutoff of 12Å for pairwise distances. This ensures that the compared models have the same number of learnable parameters, i.e. $\approx 9.4$M each. We train all models for at most 1000 epochs with a batch size of 64.

**GEOM-Drugs**  On GEOM-DRUGS, we use 10 layers of message passing for EDM*, while the variants of END feature 5 layers of message-passing in $F_\varphi$ and 5 in $\hat{x}_\theta$. The hidden size of the invariant and equivariant features is set to 192, along with an RBF expansion of dimension 64 with a cutoff of 30Å for pairwise distances (as to ensure full-connectivity). Each model features $\approx 5.4$M learnable parameters. We train all models for 10 epochs with an effective batch size of 64.

### A.6.4   Conditional generation

We use 10 layers of message passing for EDM*, while the variants of END feature 5 layers of message-passing in $F_\varphi$ and 5 in $\hat{x}_\theta$. The hidden size of the invariant and equivariant features is set to 192 , along with an RBF expansion of dimension 64 with a cutoff of 10Å for pairwise distances. We train all models for 1000 epochs with a batch size of 64.

After an initial encoding, the conditional information is introduced at the end of each message passing step, and alters the scalar hidden states through a one-layer MLP, that shares the same dimension as the hidden scalar state.

**Composition-conditioned generation**  The encoding of the condition follows that of Gebauer et al. (2022). Each atom type gets its own embedding (of dimension 64), weighted by the proportion it represents in the provided formula. The weighted embeddings of all atom types are then concatenated and flattened, and the obtained vector (of dimension $64 \times$ number of atom types ) is processed through a 2-layer MLP with 64 hidden units.

The compositions used at sampling time are extracted from the validation and test sets. For each unique formula, the model gets to generate 10 samples. The reported matching rate refers to the % of generated samples featuring the prompted composition.

**Substructure-conditioned generation**  The encoding of the condition follows that of Bao et al. (2023). Each molecule is first converted to an OPENBABEL object (O'Boyle et al., 2011) (solely based on positions and atom types), from which a fingerprint is in turn calculated. The obtained 1024-dimensional fingerprint is simply processed by a $2-$layer MLP with hidden dimensions $[512, 256]$, and a final linear projection to 192, i.e. the hidden size of the invariant features.

The model is evaluated by computing the Tanimoto similarity between the fingerprints obtained from the generated samples and the fingerprints provided as conditional inputs.

### A.7   Compute resources

All experiments were run on a single GPU. The experiments on QM9 were run on a NVIDIA SM3090 with 24 GB of memory. The experiments on GEOM-DRUGS were run on NVIDIA A100 with 40 GB of memory. Training took up to 7 days.

**Sampling**  The current implementation of END leads to $\approx$ 3x increase relative to EDM (with comparable number of learnable parameters) per function evaluation when performing sampling.

However, END usually requires much fewer number of function evaluations to achieve comparable (or better) accuracy, and we note that alternative parameterizations of the reverse process are possible. In particular, the drift of the reverse process $\hat{f}_{\theta,\varphi}$ could be learned without direct dependence on $f_\varphi$, thereby leading to very limited overhead with respect to vanilla diffusion models for sampling – only one neural network forward network would then be required. For reference, we report sampling times on QM9 (1024 samples) in Table 11, for varying numbers of integration steps.

Table 11: Average sampling time in seconds for 1024 samples on QM9. The current implementation of END leads to $\approx 3x$ increase relative to EDM.

| Model | Steps | sampling time [s] |
|-------|-------|-------------------|
| EDM | 50 | 30.4 |
| | 100 | 60.6 |
| | 250 | 149.7 |
| | 500 | 297.7 |
| | 1000 | 593.8 |
| END | 50 | 88.6 |
| | 100 | 179.6 |
| | 250 | 445.9 |
| | 500 | 886.4 |
| | 1000 | 1765.8 |

**Training** A training step on QM9 (common batch size of 64) takes on average $\approx 0.37s$ (END) vs 0.16s (EDM), this corresponds to a $\approx 2.3x$ relative increase. We observe the same trend on GEOM-Drugs (with an effective batch size of 64), a training step takes on average $\approx 0.40s$ for END vs. $\approx 0.15s$ for EDM (corresponding to a $\approx 2.7x$ relative increase). In summary, other things equal, END leads to $\approx 2.5x$ increase relative to EDM per training step while we find it to converge with a similar number of epochs. As for sampling, a direct parametrization of $\hat{f}_{\theta,\varphi}$ would enable faster training – while still requiring the evaluation of $f_\varphi$ to obtain the target reverse drift term in Eq. (5).

