# OpenReview forum: "Equivariant Neural Diffusion for Molecule Generation"
_NeurIPS.cc/2024/Conference — NeurIPS 2024 poster_

### Official Review · Reviewer_5dRq · 2024-07-05

**Soundness:** 3
**Presentation:** 3
**Contribution:** 3
**Rating:** 6
**Confidence:** 3

**Summary:**

This paper presents Equivariant Neural Diffusion (END), a novel diffusion model for 3D molecule generation. The major novelty of END over previous molecule diffusion models lies in adopting a learnable forward process based on neural flow diffusion models. Experiments show that END can achieve good performance on benchmark datasets.

**Strengths:**

- Successfully incorporate neural flow diffusion models into the equivariant 3D molecule generation framework and demonstrate its usefulness in experiments.
- Generally good, clear and well-organized writing.

**Weaknesses:**

- The novelty contribution of the proposed END method is not high, as END is largely a combination of EDM [1] and neural flow diffusion models. Particularly, the paper does not give a clear discussion or analysis about why adopting learnable forward diffusion process is useful and beneficial to 3D molecule generation, or what molecular structures can be additionally captured by END through learnable forward diffusion process compared with previous diffusion models.
- There already exist some SDE based 3D molecule generation methods like EDM-BRIDGE [2] and EEGSDE [3]. Authors are encouraged to highlight the key difference in the diffusion process between END and these methods.
- Compared with GEOLDM, END does not show better performance (Table 1 and 2), which weakens the claim about the advantages of using learnable forward process. Since the main evaluation metrics proposed by EDM [1] in 3D molecule generation are saturating in recent literatures, authors are encouraged to adopt metrics proposed by HierDiff [4] to better evaluate the quality of generated 3D molecules.

[1] Equivariant Diffusion for Molecule Generation in 3D. ICML 2022.
[2] Diffusion-based Molecule Generation with Informative Prior Bridges. NeurIPS 2022.
[3] Equivariant Energy-Guided SDE for Inverse Molecular Design. ICLR 2023.
[4] Coarse-to-Fine: a Hierarchical Diffusion Model for Molecule Generation in 3D. ICML 2023.

-------------------Post Rebuttal---------------------

I appreciate authors' efforts in addressing my concerns and questions in rebuttal. After reading over authors' rebuttal responses and pdf, I think all my concerns have been addressed so I increased my score. I hope authors will carefully add all rebuttal updates (discussion, analysis and experiment results) to the revised version of paper in the future.

**Questions:**

See Weaknesses part.

**Limitations:**

Yes.

---

> ### Author Rebuttal · Authors · 2024-08-07
>
> We thank the reviewer for their constructive feedback.
> We address the reviewer’s concerns/questions here below:
>
> **Limited technical novelty**
> As mentioned in our general rebuttal, END is indeed a combination of existing ideas. We however want to emphasize that (1) our work is the first to seek improvement by adding a learnable forward process, orthogonally to most previous work that has mostly focused on the reverse process, (2) designing an appropriate transformation was not straightforward, it required some care as to preserve the desired invariance of the learned distribution.
>
> **Relevance of learnable forward process**
> Most improvements to diffusion models for molecules have focused on designing more expressive denoisers, better noise schedules, or embedding discrete features in a continuous latent space. In END, we seek improvement in an orthogonal direction. Adding a learnable forward allows the hierarchy of latent variables to be learnt, instead of fixed as in classical diffusion models, thereby offering greater flexibility and enhanced generative modeling. We note that END can be combined with previous developments for even better performance.
>
> This enhanced generative modelling is greatly showcased on the more challenging and realistic GEOM-Drugs, where the learnable forward greatly improves connectivity (\~20% across all sampling schemes), and leads to 3D structures of better quality as evidenced by the significantly lower strain energies.
>
> Finally, the addition of a learnable forward process comes with two practically relevant by-products: (1) END achieves better performance with fewer integration steps, and (2) the learnable forward yields an improved conditional generative process – as demonstrated on two conditional tasks where cEND even outperforms  by a significant margin a model leveraging an auxiliary model through classifier-guidance (EEGSDE) (Tables 3 and 4 in the submitted manuscript).
>
> **Previous work**
> We first note that both papers were already included as baselines / previous work in the submitted manuscript, but, as suggested by the reviewer, we will add a more thorough discussion to the final version. We summarize it here:
>
> * EEGSDE is a continuous-time formulation of EDM, where conditional generation is performed by combining (1) a conditional model score model, (2) a method similar to classifier-guidance (requiring the training of an auxiliary model). In cEND, we instead only learn a conditional model, akin to classifier-free guidance.
> * Similar to END, EDM-BRIDGE builds on the observation that there exists an infinity of processes mapping from prior to target distributions. EDM-Bridge constructs one such process that incorporates some prior knowledge, i.e. part of the drift term is a physically-inspired force term. END can be seen as a generalization of EDM-Bridge, where the forward drift term is now learned instead of pre-specified. Through experiments, we show that a learnable forward performs better than a fixed one, even when the latter is physics-inspired.
>
> **Additional metrics**
> We thank the reviewer for their suggestion. We added a citation to HierDiff \[1\] and computed additional metrics taken from the paper, namely: SAScore, QED, logP and MW (see rebuttal PDF). To further assess the quality of the generated 3D structures, we also added a metric that measures the strain energy – expressed as an energy difference between the generated geometry and its relaxation (obtained through force-field geometry optimization).
>
> These additional metrics were computed on the valid x connected samples generated by each method. We converted the generated samples to SDF using OpenBabel, and read them in using RDKit. For the SAScore, we normalized the values between 0 and 1, with 0 being “difficult to synthesize” and 1 “easy to synthesize”.
> All the additional metrics are provided in the rebuttal PDF. We summarize here the main results.
>
> * On QM9, END shows better agreement with the data distribution (except for QED which is captured perfectly by all methods). In particular, the reduction in strain energy demonstrates that END yields better geometries than the baselines (Table 1 in rebuttal PDF).
> * On GEOM-Drugs, in addition to greatly improved connectivity, the SAScore, QED and logP are shown to be in better agreement with the data distribution (Table 3 in rebuttal PDF). The geometries are also shown to be of better quality as per the significantly reduced strain energy (Table 2 in rebuttal PDF).
>
>
> **Comparison to GeoLDM**
> We downloaded the GEOM-Drugs checkpoint available with the official implementation of GeoLDM, and evaluated the generated samples. We added one line with the obtained results in Table 2 in the rebuttal PDF, and summarize the findings here.
> While all samples generated by GeoLDM were effectively deemed “valid”, only around **46%** were connected, against nearly **83%** for END. The strain energy of the samples generated by END was also significantly lower, 55 kcal/mol vs 133 kcal/mol for GeoLDM, underscoring the advantages of a learnable forward process.
>
> \[1\] Coarse-to-Fine: a Hierarchical Diffusion Model for Molecule Generation in 3D. ICML 2023\.

---

> > ### Comment · Reviewer_5dRq · 2024-08-11
> > **Follow-up Response**
> >
> > I appreciate authors' efforts in addressing my concerns and questions in rebuttal. After reading over authors' rebuttal responses and pdf, I think all my concerns have been addressed so I increased my score. I hope authors will carefully add all rebuttal updates (discussion, analysis and experiment results) to the revised version of paper in the future.

---

> > > ### Author Response · Authors · 2024-08-12
> > > **Follow-up**
> > >
> > > We want to thank the reviewer for re-evaluating our submission, and providing a positively updated score.
> > > We will make sure to include all rebuttal updates in the final version of the manuscript.

---

### Official Review · Reviewer_jNQ2 · 2024-07-13

**Soundness:** 3
**Presentation:** 3
**Contribution:** 3
**Rating:** 6
**Confidence:** 3

**Summary:**

Paper presents, END, a diffusion models for 3D molecule generation that

- is equivariant to euclidean transformations and

- includes a learnable forward process.

 Specifically, the forward process in the presented models, is defined as a learnable transformation, dependent on both time and data such that the resulting latent representation $z_t$ transforms covariantly with the injected noise. This is the main difference between the forward pass of Neural Flow Diffusion Models and the proposed method.

**Strengths:**

- Can be used for both conditional and conditional molecules generation.

- Improves on existing equivariant diffusion models.

- Experimentally, the proposed method shows improvements on both conditional and unconditional generation on the QM9 and GEOM-DRUGS datasets.

**Weaknesses:**

- Lack of a thorough ablation of the proposed method.

**Questions:**

- The only form of ablation i see for the proposed model is in Table 1 where two versions of END are provided. However even here, it seems the END with $\mu_\phi$ performs similar or sometimes better (according to the presented metrics) than the full END model. Is the same pattern observed for the conditional generation tasks?

- Relating to the first question above, can a much more thorough ablation of the proposed method be provided in both testing scenarios to really ascertain the utility of the components in the proposed method?

- Can the training times for the benchmarked methods be provided for comparison? While it is stated passingly that END requires more training time, can this be cast in contrast with the baselines by providing the actual numbers?

**Limitations:**

Limitations are adequately discussed by the authors.

---

> ### Author Rebuttal · Authors · 2024-08-07
>
> We thank the reviewer for their constructive feedback. We are happy that our work was positively received by the reviewer.
> We address the reviewer’s concerns/questions here below:
>
> **Ablation**
> The key component in our method is the learnable forward process. Hence, the logical ablation is whether to include a learnable forward (=END), or not (=EDM).
>
> As stated in our global rebuttal, we made sure that all baselines shared the same amount of parameters as the models featuring a learnable forward to ensure that the difference in performance did not come from an increase in parameters. We did so by having 10 rounds of message-passing in EDM, against 5 in the forward and 5 in the reverse for END – all other things equal.
>
> As pointed out by the reviewer, the ablated version where only the mean is learned (the standard deviation of the conditional marginal is pre-specified and derived from the same noise schedule as EDM) is shown to perform on par with the full version. We believe this is due to the relative simplicity of the task. We are currently running the same ablation on the more challenging GEOM-Drugs, and hope to be able to report the results before the end of the discussion period.
>
> As nicely suggested by the reviewer, we conducted the same ablation in the conditional setting (composition-conditioned generation). The corresponding results are provided in the table here below – we will also add it to the final version. The ablated model is presented in the last row.
>
> | Matching [%] (↑) | 50   | 100  | 250  | 500  | 1000 |
> |------------------|------|------|------|------|------|
> | cEDM             | 69.6 | 73.0 | 74.1 | 76.2 | 75.5 |
> | cEND             | 89.2 | 90.1 | 91.2 | 91.5 | 91.0 |
> | cEND (mean only) | 75.7 | 79.9 | 82.7 | 83.0 | 83.5 |
>
> Regarding additional ablations, disabling equivariance in $F_\varphi$ could also be a possible option, previous work \[1\] has however clearly shown that an architecture obeying the relevant symmetries is a useful inductive bias. We would be happy to hear the additional ablations that the reviewer would like to see in the final version of the paper.
>
> **Timing**
> In terms of timing, a training step on QM9 (common batch size of 64\) takes on average \~0.37s (END) vs \~0.16s (EDM), this corresponds to a \~2.3x relative increase.
> We observe the same trend on GEOM-Drugs (with an effective batch size of 64), a training step takes on average  \~0.40s for END vs. \~0.15s for EDM (corresponding to a \~2.7x relative increase).
> We do not observe that END requires a larger number of epochs to converge, compared to EDM. All models were trained for the same number of epochs / steps.
>
> With our current implementation, sampling with END requires slightly less than 3x the time required by EDM on average. We report sampling times on QM9 (1024 samples) as an example, for varying numbers of integration steps.
>
> | time \[s\] (↑) | 50   | 100   | 250   | 500   | 1000   |
> |--------------|------|-------|-------|-------|--------|
> | EDM          | 30.4 | 60.6  | 149.7 | 297.7 | 593.7  |
> | END          | 88.6 | 179.6 | 445.9 | 886.4 | 1765.8 |
>
> However, we want to stress that END can achieve comparable (or better) accuracy with less integration steps. A concrete example of this can be seen in Table 2 in the rebuttal PDF (GEOM-Drugs), where with only 100 steps END yields better samples (connectivity and strain energy) than EDM with 1000 integration steps, i.e. in \~⅓ of the time.
> As mentioned in the global rebuttal, we note that alternative parameterizations of the reverse process are possible. In particular, the drift of the reverse process $\\hat{f}\_{\\theta, \\varphi}$ could be learned without direct dependence on $f\_{\\varphi}$, thereby leading to very limited overhead with respect to vanilla diffusion models for sampling.
>
> \[1\] Equivariant Diffusion for Molecule Generation in 3D. ICML 2022\.

---

> > ### Author Response · Authors · 2024-08-13
> > **Follow up ablation on GEOM-Drugs**
> >
> > We want to thank again the reviewer for suggesting us to run additional ablations.
> >
> > As promised in our initial rebuttal, we provide here additional results on the more challenging GEOM-Drugs -- i.e. an ablated model where only the mean is learned (the standard deviation of the conditional marginal is pre-specified and derived from the same noise schedule as EDM).
> >
> > The results are collected in the last row of the table herebelow, and presented with the initial results for better readability.
> > Similarly to the conditional setting, learning only the mean provides a clear improvement compared to the baseline, but is shown to perform slightly worse to the full model across all metrics except validity.
> >
> >
> >
> > |                 |       | At. Stab. [\%]                      | V [\%]                      | V$\times$C [\%]             | TV$_A$ [$10^{-2}$]               |
> > |-----------------|-------|-----------------------------|-----------------------------|-----------------------------|-----------------------------|
> > | Model           | Steps |                             |                             |                             |                             |
> > | EDM             | 50    | $84.7_{\scriptstyle \pm.0}$ | $93.6_{\scriptstyle \pm.2}$ | $46.6_{\scriptstyle \pm.3}$ | $10.5_{\scriptstyle \pm.1}$ |
> > |                 | 100   | $85.2_{\scriptstyle \pm.1}$ | $93.8_{\scriptstyle \pm.3}$ | $56.2_{\scriptstyle \pm.4}$ | $8.0_{\scriptstyle \pm.1}$  |
> > |                 | 250   | $85.4_{\scriptstyle \pm.0}$ | $94.2_{\scriptstyle \pm.1}$ | $61.4_{\scriptstyle \pm.6}$ | $6.7_{\scriptstyle \pm.1}$  |
> > |                 | 500   | $85.4_{\scriptstyle \pm.0}$ | $94.3_{\scriptstyle \pm.2}$ | $63.4_{\scriptstyle \pm.1}$ | $6.4_{\scriptstyle \pm.1}$  |
> > |                 | 1000  | $85.3_{\scriptstyle \pm.1}$ | $94.4_{\scriptstyle \pm.1}$ | $64.2_{\scriptstyle \pm.6}$ | $6.2_{\scriptstyle \pm.0}$  |
> > | END  | 50    | $87.1_{\scriptstyle \pm.1}$ | $84.6_{\scriptstyle \pm.5}$ | $68.6_{\scriptstyle \pm.4}$ | $5.9_{\scriptstyle \pm.1}$  |
> > |                 | 100   | $87.2_{\scriptstyle \pm.1}$ | $87.0_{\scriptstyle \pm.2}$ | $76.7_{\scriptstyle \pm.5}$ | $4.5_{\scriptstyle \pm.1}$  |
> > |                 | 250   | $87.1_{\scriptstyle \pm.1}$ | $88.5_{\scriptstyle \pm.2}$ | $80.7_{\scriptstyle \pm.6}$ | $3.5_{\scriptstyle \pm.0}$  |
> > |                 | 500   | $87.0_{\scriptstyle \pm.0}$ | $88.8_{\scriptstyle \pm.3}$ | $81.7_{\scriptstyle \pm.4}$ | $3.3_{\scriptstyle \pm.0}$  |
> > |                 | 1000  | $87.0_{\scriptstyle \pm.0}$ | $89.2_{\scriptstyle \pm.3}$ | $82.5_{\scriptstyle \pm.3}$ | $3.0_{\scriptstyle \pm.0}$  |
> > | **END (mean only)** | 50    | $85.6_{\scriptstyle \pm.1}$ | $87.8_{\scriptstyle \pm.2}$ | $66.0_{\scriptstyle \pm.4}$ | $7.9_{\scriptstyle \pm.0}$  |
> > |                 | 100   | $85.8_{\scriptstyle \pm.1}$ | $89.9_{\scriptstyle \pm.1}$ | $73.7_{\scriptstyle \pm.4}$ | $6.1_{\scriptstyle \pm.1}$  |
> > |                 | 250   | $85.7_{\scriptstyle \pm.1}$ | $91.2_{\scriptstyle \pm.2}$ | $77.4_{\scriptstyle \pm.4}$ | $5.0_{\scriptstyle \pm.1}$  |
> > |                 | 500   | $85.8_{\scriptstyle \pm.1}$ | $91.6_{\scriptstyle \pm.1}$ | $78.6_{\scriptstyle \pm.3}$ | $4.8_{\scriptstyle \pm.1}$  |
> > |                 | 1000  | $85.8_{\scriptstyle \pm.1}$ | $91.8_{\scriptstyle \pm.1}$ | $79.4_{\scriptstyle \pm.4}$ | $4.6_{\scriptstyle \pm.0}$|

---

### Official Review · Reviewer_K86T · 2024-07-13

**Soundness:** 3
**Presentation:** 3
**Contribution:** 3
**Rating:** 7
**Confidence:** 4

**Summary:**

The Equivariant Neural Diffusion (END) model is a novel approach for molecule generation in 3D that maintains equivariance to Euclidean transformations. Unlike traditional diffusion models that use a pre-specified forward process, END introduces a learnable forward process, parameterized through a time- and data-dependent transformation. This innovation allows the model to adapt better to the underlying data distribution. Experimental results demonstrate that END outperforms strong baselines on standard benchmarks for both unconditional and conditional generation tasks, particularly excelling in generating molecules with specific compositions and substructures. This flexibility in modeling complex molecular structures suggests significant potential for applications in drug discovery and materials design.

**Strengths:**

**Originality:** END introduces a novel learnable forward process, diverging from the fixed processes in traditional diffusion models. This allows the model to better adapt to the underlying data distribution, especially in complex 3D molecular structures. It creatively combines elements from Neural Function Matching Diffusion Models (NFDM) and Equivariant Diffusion Models (EDM), enhancing the generative process while maintaining E(3) equivariance.

**Quality:** The model demonstrates superior performance in generating 3D molecular structures, outperforming strong baselines in both unconditional and conditional settings. It excels in generating stable, valid, and unique molecules, particularly evident in its results on the GEOM-DRUGS dataset. Comprehensive experiments and ablation studies confirm the robustness and reliability of the model.

**Clarity:** The paper provides a detailed and clear exposition of the methodology, including the formulation of the learnable forward process, parameterization, and evaluation metrics. The inclusion of algorithmic steps and extensive experimental details facilitates replicability for researchers.

**Significance:** END represents a substantial advancement in generative modeling for 3D molecules, addressing limitations of prior models by improving sample quality and generation speed. Its ability to maintain equivariance while achieving high performance has significant implications for applications in drug discovery and materials design, potentially transforming these fields.

**Weaknesses:**

**Performance Consistency:** The performance of END is not consistently superior to existing baselines. Although it shows competitive results, there are instances where traditional models, like EDM and its variants, outperform END, particularly in metrics such as validity and uniqueness across different datasets.

**Complexity and Scalability:** The added complexity of a learnable forward process, while innovative, increases the model’s training time and resource requirements. END requires more computational resources and longer training periods compared to simpler, fixed-process models like EDM . The model operates on fully-connected graphs, which limits its scalability to larger datasets and more complex molecular structures. This constraint can hinder its applicability in more demanding real-world scenarios.

**Questions:**

- Why was GeoDiff not included as a baseline in your comparisons?
- Could you provide more detailed ablation studies that isolate the contributions of the learnable forward process and other key components of END?

**Limitations:**

**Computational Complexity:** The learnable forward process in the END model increases computational complexity, resulting in longer training times and higher resource usage. The authors should discuss potential optimizations to reduce this overhead, such as more efficient algorithms or hybrid approaches. Comparing training times and resource requirements with simpler models would provide useful insights into the model's efficiency.

**Scalability Issues:** END's architecture limits its scalability to larger datasets and more complex molecular structures like proteins. To enhance scalability, the authors should explore strategies like sparse representations or hierarchical approaches. These methods could enable the model to handle larger and more complex datasets effectively.

---

> ### Author Rebuttal · Authors · 2024-08-07
>
> We thank the reviewer for their constructive feedback. We are happy that our work was positively received by the reviewer.
> We address the reviewer’s concerns/questions here below.
>
> **Performance Consistency**
> Regarding validity, we first note that (1) cheminformatics software implicitly adds hydrogens if an atom has a valence smaller than expected, (2) validity is usually computed on the largest connected fragment (thereby ignoring the remaining atoms).
> On QM9, mol. stability gives a better picture, as it ensures that each atom exactly has the desired variance. Models including a learnable forward pass are shown to perform significantly better than baselines without. Improvements are also observed across other metrics (Table 1 in submission).
> On the more challenging GEOM-Drugs, we believe that evaluating the model in terms of (validity x connectivity) is more realistic. According to that metric, END is clearly better than the baseline and samples far more connected molecules (Table 2 in rebuttal PDF).
>
> **GeoDiff**
> To the best of our knowledge, GeoDiff \[1\] is a diffusion model for conformer generation, i.e. generation of coordinates given a molecular graph. Perhaps the reviewer had another model in mind?
>
> **Ablation**
> The key component in our method is the learnable forward process. Hence, the logical ablation is whether to include a learnable forward (=END), or not (=EDM).
>
> To ensure that the difference in performance does not come from an increase in parameters, we made sure that all EDM baselines shared the same amount of parameters as the models featuring a learnable forward. We did so by having 10 rounds of message-passing in EDM, against 5 in the forward and 5 in the reverse for END – all other things equal.
>
> We additionally provided another ablated version of END on QM9, where only the mean is learned, whereas the standard deviation of the conditional marginal is pre-specified and derived from the same noise schedule as EDM.
> As suggested by another reviewer **\[jNQ2\]**, we conducted the same experiment in the conditional setting, where fixing the standard deviation is shown to lead to a small decrease in performance.
> We are currently running the same ablation on GEOM-Drugs, and hope to be able to report the results before the end of the discussion period.
>
> **Complexity and Scalability**
> **Complexity:**
> In terms of training, we did not observe that END required more epochs than the baseline to reach convergence. However, it is correct that each training step takes longer, and that the total training time is increased. All other things equal, we evaluated that increase to \~2.5x relative to EDM. In terms of resources, END could be trained on a single GPU for both datasets.
>
> Regarding sampling, our current implementation incurs a \~3x increase relative to EDM.
> However, we emphasize that END can be competitive with fewer integration steps. As a concrete example, on GEOM-Drugs, END with only 100 steps yields better samples than EDM with 1000 integration steps, i.e. in \~⅓ of the time.
> Finally, alternative efficient parameterizations of the reverse process are possible.  In particular, the drift of the reverse process $\\hat{f}\_{\\theta, \\varphi}(\\boldsymbol{z}\_t, t)$ could be learned without direct dependence on $f\_{\\varphi}$, thereby leading to a very limited overhead with respect to vanilla diffusion models for sampling.
>
> **Scalability:** As all concurrent approaches, scaling to large systems is currently limited by the full connectivity of the message passing scheme.
> An element specific to END, is that, due to the learnable forward, the prior is no longer forced to be $\\mathcal{N}(0, I)$ as in conventional diffusion models and can instead e.g. become size-specific. For large molecules, one could imagine scaling the prior with the number of atoms present in the system, i.e. $\\mathcal{N}(0, sI)$ where $s$ is a function of the number of atoms. Combined with a distance-based cutoff function to determine the neighborhood, this would allow for a more graceful scaling – as early steps of the denoising process would not result in fully-connected message passing as it is effectively the case with a simple $\\mathcal{N}(0, I)$ prior.
>
> Another possibility to scale to systems such as large coordination complexes, is to design priors that “encode shapes”, e.g. square planar geometry could be built as a center and 4 groups of atoms representing the coordinated ligands, and thereby enabling more local message passing schemes.
>
> \[1\] GeoDiff: a Geometric Diffusion Model for Molecular Conformation Generation, ICLR 2022

---

### Official Review · Reviewer_hMME · 2024-07-13

**Soundness:** 2
**Presentation:** 2
**Contribution:** 2
**Rating:** 4
**Confidence:** 4

**Summary:**

This paper proposes an extension of diffusion models dubbed Equivariant Neural Diffusion, which leverages  a learnable forward diffusion process to enhance flexibility. The entire framework has been constructed such that physical symmetry, i.e., equivariance/invariance, of the density is preserved. Experiments are performed on QM9 and DRUGS datasets in the task of molecule generation from scratch as well as controllable generation.

**Strengths:**

1. The presentation is mostly clear and the method is easy to follow.
2. The method has been demonstrated to perform favorably in the controllable generation setting, even when the number of sampling steps is limited to, e.g., 50.

**Weaknesses:**

1. The proposed approach seems to be an incremental combination of geometric diffusion models and neural flow diffusion models. The way of combining these two flavors incurs limited technical novelty. The core design mostly lies in constructing $F\_\varphi$ which is equivariant.

2. The performance on QM9 and DRUGS is a bit marginal compared with the selected baselines.

3. Missing important baselines, e.g., GeoBFN [1], which has shown strong performance on the same task, i.e., molecule generation. Moreover, GeoBFN can also achieve high performance with very few sampling steps, even only 20.

[1] Song et al. Unified generative modeling of 3d molecules with bayesian flow networks. In ICLR'24.

**Questions:**

Q1. How does the method perform compared with GeoBFN, especially with different number of sampling steps?

Q2. Could the method be applied to systems with larger scales, e.g., proteins?

**Limitations:**

The authors have discussed the limitations, including scaling, limited application scope, etc.

---

> ### Author Rebuttal · Authors · 2024-08-07
>
> We thank the reviewer for their constructive feedback.  We address the reviewer’s concerns/questions here below:
>
> **Limited technical novelty**
> As mentioned in our general rebuttal, END is indeed a combination of existing ideas. We however want to emphasize that (1) our work is the first to seek improvement by adding a learnable forward process, orthogonally to most previous work that has mostly focused on the reverse process, (2) designing an appropriate transformation was not straightforward, it required some care as to preserve the desired invariance of the learned distribution.
>
> **Marginal performance gain**
> While we agree with the reviewer that the performance gains are limited on QM9, we have shown that END works considerably better than the considered baselines on GEOM-Drugs -- as summarized in the general rebuttal. Most notably, END significantly improves connectivity and geometry quality. It also shows better agreement with the training data on (newly added) drug-related metrics (SAScore, QED and logP).
>
> **Missing baseline GeoBFN**
> Unfortunately, since GeoBFN was published on arXiv \~2 months prior to the submission deadline a thorough comparison to this concurrent work is unfeasible. However, we extracted the relevant numbers presented in that paper and a direct comparison with GeoBFN in Tables 1 and 2 will be part of the final version. We summarize the comparison below.
>
> On QM9, END generally performs on par with GeoBFN on the stability / validity metrics, across sampling steps. We could not evaluate the additional metrics as we do not have access to samples / pretrained models for GeoBFN.
>
> On the more realistic GEOM-Drugs, END is shown to outperform GeoBFN in terms of atom stability (first row in the table below) – especially for small numbers of steps. As we can not check whether GeoBFN generates disconnected fragments, solely based on validity GeoBFN seems to perform better (second row in the table below).
>
> |        | 50   | 100  | 500  | 1000 |
> |--------|------|------|------|------|
> | END    | 87.1 | 87.2 | 87.0 | 87.0 |
> | GeoBFN | 75.1 | 78.9 | 81.4 | 85.6 |
> |        |      |      |      |      |
> | END    | 84.6 | 87.0 | 88.5 | 88.8 |
> | GeoBFN | 91.7 | 93.1 | 93.5 | 92.1 |
>
> It would be interesting to evaluate the connectivity / strain energy of the samples produced by GeoBFN for a full comparison.
>
> **Application to large systems**
> There is no restriction to the systems the END could be applied to.
> As all concurrent approaches, scaling to large systems is currently limited by the full connectivity of the message passing scheme.
> As for other methods treating categorical features as a continuous relaxation, we also note that for systems with a large number of atom types, alternatives to one-hot encoding (linear scaling wrt cardinality of space) might be beneficial. Methods such as Analog Bits \[1\] (logarithmic scaling), or continuous lower-dimensional embedding such as that of GeoLDM \[2\] are interesting approaches to deal with discrete features within END.
>
> \[1\] Analog Bits: Generating Discrete Data using Diffusion Models with Self-Conditioning, ICLR 2023\.
> \[2\] Geometric Latent Diffusion Models for 3D Molecule Generation, ICML 2023

---

> > ### Author Response · Authors · 2024-08-12
> > **Follow up**
> >
> > Again, we want to thank the reviewer for providing constructive feedback.
> >
> > We believe that we have now addressed the weaknesses raised in the initial review.
> >
> > We are happy to clarify any issue that should remain.

---

> ### Comment · Reviewer_hMME · 2024-08-12
>
> Thank you for the response. However I do believe the paper would be in better shape by additional efforts in some modifications e.g., adding comparison with advanced baselines (e.g., GeoBFN). I understand that GeoBFN was published ~2months prior to the deadline but the whole experiment would just take around several days and a fair comparison is expected, since you both work on the same dataset and benchmark and claim the advantage of fewer step sampling. Moreover, the work still seems technically incremental. I will slightly increase the score but still hold an opinion that the paper could be further strengthened by either showcasing strong performance against sota methods or exploring novel applications/use cases that constitutes a unique contribution.

---

> > ### Author Response · Authors · 2024-08-13
> > **Follow up**
> >
> > We thank the reviewer for engaging in the discussion, and helping us improve the paper further.
> >
> > First, we want to reiterate that a comparison with GeoBFN based on the published results is provided in our rebuttals, and will be included in the final version of the paper — details are provided below for reference.
> >
> > Second, we want to emphasize that END showcases strong performance against SOTA methods, and specifically against GeoBFN.
> > On QM9, END demonstrates a level of performance very similar to that of GeoBFN (see below).
> > On GEOM-Drugs, END clearly outperforms GeoBFN in terms of atom stability (e.g. END’s 50-step atom stability is better than GeoBFN’s 1000-step), while GeoBFN does slightly better in terms of validity. We note that GeoLDM, a very relevant baseline, is shown to lead higher validity than both GeoBFN and END, but that connectivity and geometry quality is subpar compared to END — highlighting that concluding anything from that metric alone is difficult.
> >
> > Finally, as suggested by the reviewer, we will run and collect additional metrics for GeoBFN, and include them in the final version of the paper. However, due to the unavailability of public checkpoints, we can unfortunately not perform that experiment before the discussion period ends.
> >
> >
> >
> >
> > **QM9**
> > | Metrics / Steps |        | 50              | 100            | 500            | 1000            |
> > |-----------------|--------|-----------------|----------------|----------------|-----------------|
> > | At. Sta.        | END    | $98.6 \pm 0.0$  | $98.8 \pm 0.0$ | $98.9 \pm 0.0$ | $98.9 \pm 0.0$  |
> > |                 | GeoBFN | $98.28\pm 0.1$  | $98.64\pm 0.1$ | $98.78\pm 0.8$ | $99.08\pm 0.06$ |
> > |                 |        |                 |                |                |                 |
> > | Mol. Sta.       | END    | $84.6 \pm 0.1$  | $87.4 \pm 0.2$ | $88.8 \pm 0.4$ | $89.1 \pm 0.1$  |
> > |                 | GeoBFN | $85.11 \pm 0.5$ | $87.21\pm 0.3$ | $88.42\pm 0.2$ | $90.87\pm 0.2$  |
> > |                 |        |                 |                |                |                 |
> > | V               | END    | $92.7\pm 0.1$   | $94.1\pm 0.0$  | $94.8\pm 0.2$  | $94.8\pm 0.1$   |
> > |                 | GeoBFN | $92.27\pm 0.4$  | $93.03\pm 0.3$ | $93.35\pm 0.2$ | $95.31\pm 0.1$  |
> > |                 |        |                 |                |                |                 |
> > | V x U           | END    | $91.4\pm 0.1$   | $92.3\pm 0.2$  | $92.8\pm 0.2$  | $92.6\pm 0.2$   |
> > |                 | GeoBFN | $90.72\pm 0.3$  | $91.53\pm 0.3$ | $91.78\pm 0.2$ | $92.96\pm 0.1$  |
> >
> >
> >
> > **GEOM-Drugs**
> > | Metrics / Steps |        | 50             | 100            | 500            | 1000           |
> > |-----------------|--------|----------------|----------------|----------------|----------------|
> > | At. Sta.        | END    | $87.1 \pm 0.1$ | $87.2 \pm 0.1$ | $87.0 \pm 0.0$ | $87.0 \pm 0.0$ |
> > |                 | GeoBFN | $75.11$        | $78.89$        | $81.39$        | $85.60$        |
> > |                 |        |                |                |                |                |
> > | V               | END    | $84.6 \pm 0.5$ | $87.0 \pm 0.2$ | $88.8 \pm 0.3$ | $89.2 \pm 0.3$ |
> > |                 | GeoBFN | $91.66$        | $93.05$        | $93.47$        | $92.08$        |

---

### Author Rebuttal · Authors · 2024-08-07

We thank the reviewers for their constructive feedback.

We are pleased that the reviewers [**hMME,5dRq,K86T**] found the presentation of END clear, its construction original **[K86T]**, that it is a potentially significant contribution to the field **[K86T],** and that it demonstrates experimental benefits over existing models in unconditional and conditional settings [**hMME,K86T,jNQ2**].

In this global rebuttal, we address the main concerns of the reviewers. We also provide a detailed answer to each reviewer below.

## Performance and metrics

Three reviewers [**hMME,5dRq,K86T**] remarked that we did not demonstrate a clear improvement over competing methods. We respectfully disagree, and to highlight the strong performance improvement of END, we have included several additional metrics. And both the original metrics and the newly included ones show that END strongly outperforms competitors.

### Additional metrics

We added 5 metrics. As specifically suggested by reviewer **5dRq**, they are taken from HierDiff [1]:

* **SAS**: Synthetic accessibility score;
* **QED**: Quantitative estimation of drug-likeness;
* **logP**: Partition coefficient;
* **MW**: Molecular weight.

We also provided a metric measuring the quality of the generated 3D geometries: **Strain Energy**, as the difference in energy when relaxing the generated molecules (using RDKit MMFF)

### QM9

On the simpler QM9 benchmark, in terms of validity, stability, and uniqueness, END reaches a performance level similar to that of current SOTA methods (GeoLDM and GeoBFN) but these metrics are essentially saturated.

More importantly END displays better agreement with the dataset distribution compared with EDM: For example, TV on atom types go from 2.5 (EDM) to 1.2 (END) at 1000 sampling steps (Table 1 in submitted manuscript). Furthermore, on all additional metrics, END shows better agreement with the data distribution (except for QED which is captured perfectly by all methods) as shown in Table 1 in rebuttal PDF. In particular, the reduction in strain energy demonstrates that END yields better geometries than the baselines.

### GEOM-Drugs

On the more challenging and realistic GEOM-Drugs we show **large improvements** (Tables 2 and 3 in rebuttal PDF):

- (i) ~20% improvement over EDM in terms of “Validity x Connectivity”;
- (ii) ~50% improvement on “TV on atom types”;
- (iii) improved 3D geometries as per the significantly lower strain energy;
- (iv) improved SAS, QED and logP.

As suggested by reviewer **5dRq**, we conducted a more extensive comparison with GeoLDM by sampling using the publicly available checkpoint.  We summarize here our findings (Table 2 in the rebuttal PDF):
* V x C improves from 45.8% (GeoLDM) to 82.5% (END)
* TV on atom types improves from 10.6 (GeoLDM) to 3.0 (END)
* Strain Energy improves from 133.5 (GeoLDM) to 55.0 (END).

We believe these results demonstrate the strong improvement on the prior SOTA. All metrics are collected in tables in the rebuttal PDF.

## Missing baselines / Previous work

**hMME** requested a comparison to the very recent GeoBFN [2] (posted on arXiv ~2 months prior to the submission). While already part of the literature review, we will add the comparison in Tables 1 and 2 in the final version.

**5dRq** requested a comparison to two previous works [3, 4]. While already included in the submission as baseline/related work, we will additionally provide a paragraph discussing the key differences in the final version (see answer to **5dRq** for summary).

## Ablations

**K86T** and **jNQ2** mentioned that more detailed ablation studies could make the paper even stronger.
First, we want to stress that we made sure that all baselines shared the same amount of parameters as the models featuring a learnable forward, as to ensure that the difference does not simply come from an increase in parameters.
Following reviewer **jNQ2**'s suggestion, we ran an additional ablation study, and added the results in our rebuttal to **jNQ2**. We are currently running an ablated version of END (mean-only) on GEOM-Drugs – we hope to be able to report the results during the discussion period.

## Timing

Two reviewers  [**K86T, jNQ2**] requested more information about the increased training/sampling time.
In summary, other things equal, END leads to ~2.5x increase relative to EDM per training step and requires the same number of epochs to converge.
Regarding sampling, our current implementation leads to ~3x increase relative to EDM per function evaluation. However, END usually requires much fewer number of function evaluations to achieve comparable (or better) accuracy and we note that alternative parameterizations of the reverse process are possible. In particular, the drift of the reverse process $\\hat{f}\_{\\theta, \\varphi}$ could be learned without direct dependence on $f\_{\\varphi}$, thereby leading to very limited overhead with respect to vanilla diffusion models for sampling. We will add these important considerations to the final version of the manuscript.

## Technical novelty

Some reviewers [**5dRq, hMME**] pointed out some limited novelty.
END is indeed a combination of existing ideas. We however want to emphasize that (1) our work is the first molecule generation diffusion model to seek improvement by adding a learnable forward process, orthogonally to most previous work that has mostly focused on the reverse process, (2) designing an appropriate transformation was not straightforward, it required some care as to preserve the desired invariance of the learned distribution.

[1] Coarse-to-Fine: a Hierarchical Diffusion Model for Molecule Generation in 3D. ICML 2023.
[2] Song et al. Unified generative modeling of 3d molecules with bayesian flow networks. In ICLR'24.
[3] Diffusion-based Molecule Generation with Informative Prior Bridges. NeurIPS 2022.
[4] Equivariant Energy-Guided SDE for Inverse Molecular Design. ICLR 2023.

---

### Decision · Program_Chairs · 2024-09-25

**Decision:**

Accept (poster)

**Comment:**

The reviewers all agreed that the method performs well (esp. in challenging conditions), and most reviewers noted that the paper is well written. The method is a novel combination of existing ideas, and different reviewers appraised this differently, with some finding it somewhat incremental (e.g. hMME) while others (e.g. K86T) finding it a creative combination. Many minor concerns raised by the reviewers were effectively addressed during the discussion, and in the end only hMME remained skeptical. Their primary concern is a comparison to the recent work GeoBFN. The authors have now shown their method to be competitive, and are working on running comparisons with additional metrics as requested.

In my view the method is sufficiently novel, and already fairly well benchmarked in the initial submission. The authors have been very active improving the paper and addressing reviewer concerns during the discussion period, and overall this now looks like a solid contribution. So I recommend the paper for acceptance at NeurIPS.